# OpenSpliceAI provides an efficient modular implementation of SpliceAI enabling easy retraining across nonhuman species

**Kuan-Hao Chao[1,2]\*[†], Alan Mao[1,2,3][†], Anqi Liu[1], Steven L Salzberg[1,2,3,4]\*, Mihaela Pertea[1,2,3]\***

[1]Department of Computer Science, Johns Hopkins University, Baltimore, United States; [2]Center for Computational Biology, Johns Hopkins University, Baltimore, United States; [3]Department of Biomedical Engineering, Johns Hopkins University, Baltimore, United States; [4]Department of Biostatistics, Johns Hopkins University, Baltimore, United States

**\*For correspondence:**
kh.chao@cs.jhu.edu (K-HC);
salzberg@jhu.edu (SLS);
mpertea@jhu.edu (MP)

[†]These authors contributed equally to this work

**Competing interest:** The authors declare that no competing interests exist.

## eLife Assessment

This **valuable** study introduces a modern and accessible PyTorch reimplementation of the widely used SpliceAI model for splice site prediction. The authors provide **convincing** evidence that their OpenSpliceAI implementation matches the performance of the original while improving usability and enabling flexible retraining across species. These advances are likely to be of broad interest to the computational genomics community.

**Abstract** The SpliceAI deep learning system is currently one of the most accurate methods for identifying splicing signals directly from DNA sequences. However, its utility is limited by its reliance on older software frameworks and human-centric training data. Here, we introduce OpenSpliceAI, a trainable, open-source version of SpliceAI implemented in PyTorch to address these challenges. OpenSpliceAI supports both training from scratch and transfer learning, enabling seamless retraining on species-specific datasets and mitigating human-centric biases. Our experiments show that it achieves faster processing speeds and lower memory usage than the original SpliceAI code, allowing large-scale analyses of extensive genomic regions on a single GPU. Additionally, OpenSpliceAI's flexible architecture makes for easier integration with established machine learning ecosystems, simplifying the development of custom splicing models for different species and applications. We demonstrate that OpenSpliceAI's output is highly concordant with SpliceAI. In silico mutagenesis analyses confirm that both models rely on similar sequence features, and calibration experiments demonstrate similar score probability estimates.

## Introduction

Predicting splice sites within primary DNA sequences has a wide range of uses, including understanding gene regulation, identifying alternative protein isoforms, and detecting sequence variants that affect splicing (*Black, 2000*; *Braunschweig et al., 2013*; *Wagner et al., 2023*; *Xiong et al., 2015*). Splicing is a complex and tightly regulated process that enables the production of multiple

protein isoforms from a single gene, contributing to cellular complexity, adaptability, and diversity across different cells and tissues (*Blencowe, 2006*; *Johnson et al., 2003*; *Wang et al., 2008*).

Aberrant splicing regulation can contribute to a wide range of diseases, including some types of cancer (*Bonnal et al., 2020*; *Jung et al., 2015*; *Lee and Abdel-Wahab, 2016*; *Supek et al., 2014*; *Sveen et al., 2016*), neurodegenerative disorders (*Li et al., 2021*; *Mills and Janitz, 2012*; *Nikom and Zheng, 2023*), cardiovascular diseases (*Gotthardt et al., 2023*; *Martí-Gómez et al., 2022*), metabolic syndromes (*Dlamini et al., 2017*; *Moore et al., 2010*), and various genetic conditions (*Segal and Widom, 2009*; *Xiong et al., 2015*). Notably, Duchenne muscular dystrophy (*Aartsma-Rus et al., 2002*; *McClorey et al., 2005*) and spinal muscular atrophy (*Burnett et al., 2009*; *Lorson et al., 1999*; *Naryshkin et al., 2014*) are well-known examples of disorders arising from splicing defects. It has been estimated that 15–50% of disease-causing mutations in humans influence splice site selection (*Baralle and Giudice, 2017*; *Barash et al., 2010*; *Wang and Cooper, 2007*), underscoring the critical need for precise modeling of splicing regulation at the DNA level and accurate interpretation of model predictions.

Building on advancements in deep learning, particularly convolutional neural networks (CNNs), genomics researchers have made substantial progress in modeling complex, long-range dependencies in DNA sequences. These models have driven significant improvements in predictive accuracy across diverse applications, including regulatory grammar (*Alipanahi et al., 2015*; *Kelley et al., 2018*; *Kelley et al., 2016*; *Zhou et al., 2018*; *Zhou and Troyanskaya, 2015*), 3D genome organization (*Fudenberg et al., 2020*), mRNA stability (*Agarwal and Kelley, 2022*), and notably, splice site prediction (*Jaganathan et al., 2019*; *Sokolova et al., 2024*). Among these, SpliceAI (*Jaganathan et al., 2019*) stands out as the leading tool for splice site prediction, applying a deep residual CNN architecture to identify patterns dictating splicing mechanisms directly from primary sequences without relying on human-engineered features.

Despite its success, SpliceAI has limitations that hinder its broader application. The official implementation relies on an outdated version of TensorFlow (*Abadi et al., 2016*) and Keras, which may not function well with newer machine learning frameworks such as PyTorch (*Paszke et al., 2019*), which has been widely adopted in recent years. Additionally, SpliceAI's use of human training data limits its performance on nonhuman species, suggesting that a retrained module could provide substantial advantages for those wishing to use it on model organisms or other species.

To address these limitations, we developed OpenSpliceAI, a trainable open-source implementation of SpliceAI in PyTorch. OpenSpliceAI supports both training from scratch and transfer-learning approaches, making it adaptable to species-specific datasets. As we show in our experiments below, OpenSpliceAI offers faster processing speed, reduced memory usage, and efficient GPU utilization, enabling analysis of long sequences and large datasets on a single GPU. In silico mutagenesis (ISM) analyses revealed the features that both SpliceAI and OpenSpliceAI rely on for making predictions. Calibration experiments showed that OpenSpliceAI models are well calibrated, improving the reliability of splice site predictions.

## Results

### OpenSpliceAI: an open-source splice site prediction framework in PyTorch

Our new system, which we call OpenSpliceAI, is a suite of modular Python scripts that provide researchers with a user-friendly computational framework to study RNA splicing. OpenSpliceAI is an open-source version of SpliceAI, a highly accurate splice site prediction method (*Jaganathan et al., 2019*). By replacing TensorFlow and Keras with the more-efficient PyTorch framework, OpenSpliceAI offers improved performance, scalability, and compatibility with current machine learning workflows (see Discussion).

The framework faithfully replicates SpliceAI's architecture while extending its functionality. It is important to note that the models produced by OpenSpliceAI and the original SpliceAI are not identical. Variations in weight initialization, data shuffling, batch normalization, and optimizer stochasticity introduce subtle differences between the models, as discussed in detail in the Discussion section. Additionally, we provide new modules for training the network, allowing for easy retraining on other species, which we show provides more accurate performance on those species. OpenSpliceAI

supports custom model training on long DNA sequences and offers both training-from-scratch and transfer-learning approaches to adapt models to species-specific datasets. We also conducted experiments to analyze the effects of DNA mutations on OpenSpliceAI's predicted scores for donor and acceptor sites and show how to use it to identify cryptic splicing events, where a mutation can activate a normally dormant splice site.

To streamline its use, OpenSpliceAI offers six subcommands for data preprocessing, model training, transfer learning, calibration, prediction, and variant analysis (see *Figure 1*; *Figure 1—figure supplements 1–3*). Detailed functionalities of each module are described in Methods.

## Training OpenSpliceAI with human MANE annotation

Using the OpenSpliceAI framework, we trained a new PyTorch version of SpliceAI using protein-coding genes annotated in the RefSeq MANE v1.3 database. MANE provides a standardized set of human gene annotations covering nearly all known protein-coding genes, with one transcript per gene, and ensures that the transcripts are represented identically in the RefSeq and Ensembl/GENCODE annotations of the GRCh38 human reference genome (*Morales et al., 2022*).

Gene sequences and splice site labels from MANE annotations were extracted and one-hot encoded into tensors for OpenSpliceAI training. Models were trained using flanking sequences of 80, 400, 2000, and 10,000 nucleotides, with five models trained for each sequence length (see *Figure 2A*). OpenSpliceAI assigns a score to each position that is an estimate of the probability that the position is a donor site, acceptor site, or neither (see *Figure 2B* and Methods). In this setup, 'OpenSpliceAI' refers to the reimplemented framework, 'OSAI' refers to the model, and 'OSAI$_{MANE}$' denotes the model trained specifically with the MANE annotation. 'SpliceAI-Keras' denotes the original published SpliceAI model, which was trained using the canonical transcripts from GENCODE version V24lift37 from the hg19/GRCh37 reference genome.

To compare the performance of OSAI$_{MANE}$ and SpliceAI-Keras, both models were evaluated on a held-out test set comprising genes from MANE annotations on GRCh38 chromosomes 1, 3, 5, 7, and 9. Paralogous genes were excluded to prevent data leakage by aligning the test set against the training sets of both models (Methods). OSAI$_{MANE}$ (*Figure 2C and D*, orange curve) showed performance comparable to SpliceAI-Keras (*Figure 2C and D*, blue curve) across metrics, including top-1 accuracy, area under the precision-recall curve (AUPRC), precision, recall, and F1 score for donor and acceptor splice sites (*Figure 2—figure supplement 1*, Methods). Results are presented as error bar plots showing the mean and ±1 standard error for each metric.

The best-performing OSAI$_{MANE}$ model, trained with 10,000 nt flanking sequences, demonstrated slight yet consistent improvements over SpliceAI-Keras across all metrics (*Figure 2*, *Figure 2—figure supplement 1*). Specifically, it achieved a top-1 accuracy increase of 1.25% for donor sites and 1.56% for acceptor sites, an F1 score gain of 1.20% for donor sites and 1.04% for acceptor sites, and an AUPRC improvement of 1.90% for donor sites and 1.76% for acceptor sites.

Performance improved with longer flanking sequences, consistent with previous SpliceAI findings. The largest gains occurred between 80 nt and 400 nt, with accuracy increasing by 62% for donor sites and 74% for acceptor sites. In comparison, improvements were smaller between 400 nt and 2000 nt (3.2% for donors and 2.5% for acceptors) and between 2000 nt and 10,000 nt (3.8% for donors and 4.2% for acceptors). Cross-species evaluations with mouse (*Figure 2—figure supplement 2*), honeybee (*Figure 2—figure supplement 3*), zebrafish (*Figure 2—figure supplement 4*), and *Arabidopsis* (*Figure 2—figure supplement 5*) test sets confirmed comparable performance across species.

OpenSpliceAI supports both targeted splice site prediction and genome-wide predictions across full chromosomes. Its 'variant' submodule enables researchers to assess the splicing impacts of specific variants, such as acceptor and donor site gains or losses, using pre-trained models. Our performance benchmarks of the 'predict' (*Figure 2E*, *Figure 2—figure supplement 6A–F*) and 'variant' (*Figure 2—figure supplement 6G–L*) submodules demonstrate that OSAI outperforms SpliceAI in processing speed, memory usage, and GPU efficiency (see Discussion).

## Retraining models with different species using OpenSpliceAI framework

To assess whether SpliceAI can generalize across different species and to demonstrate the ease of retraining models with OpenSpliceAI, we selected four model organisms representing diverse taxa:

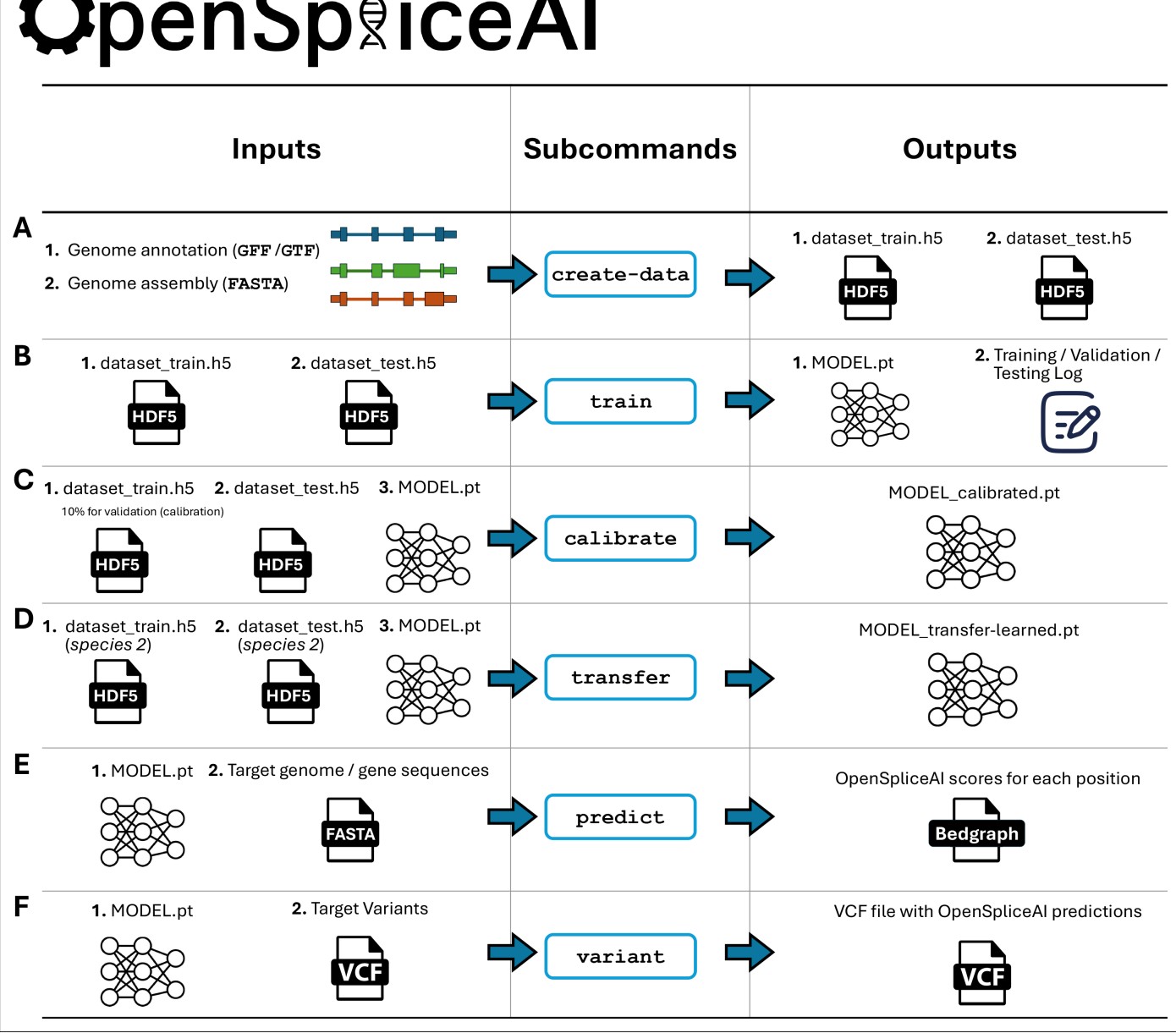

**Figure 1.** Overview of the OpenSpliceAI design. This toolkit features six primary subcommands: (**A**) The 'create-data' subcommand processes genome annotations in GFF/GTF format and genome sequences in FASTA format to produce one-hot encoded gene sequences (**X**) and corresponding labels (**Y**), both stored in HDF5 format. (**B**) The 'train' subcommand utilizes the HDF5 files generated by 'create-data' to train the SpliceAI model using PyTorch, resulting in a serialized model in PT format. This process also generates logs for training, testing, and validation. (**C**) The 'calibrate' subcommand takes both training and test datasets along with a pre-trained model in PT format. It randomly allocates 10% of the training data as a validation (calibration) set, which is then used to adjust the model's output probabilities so that they more accurately reflect the observed empirical probabilities during evaluation on the test set. (**D**) The 'transfer' subcommand allows for model customization using a dataset from a different species, requiring a pre-trained model in PT format and HDF5 files for transfer learning and testing. (**E**) The 'predict' subcommand enables users to predict splice site probabilities for sequences in given FASTA files. (**F**) The 'variant' subcommand assesses the impact of potential SNPs and indels on splice sites using VCF format files, providing predicted cryptic splice sites.

The online version of this article includes the following figure supplement(s) for figure 1:

**Figure supplement 1.** Overview of the OpenSpliceAI architectures trained with different flanking sequence lengths.

**Figure supplement 2.** Decision-making and workflow of the predict subcommand.

**Figure supplement 3.** Decision-making and workflow of the variant subcommand.

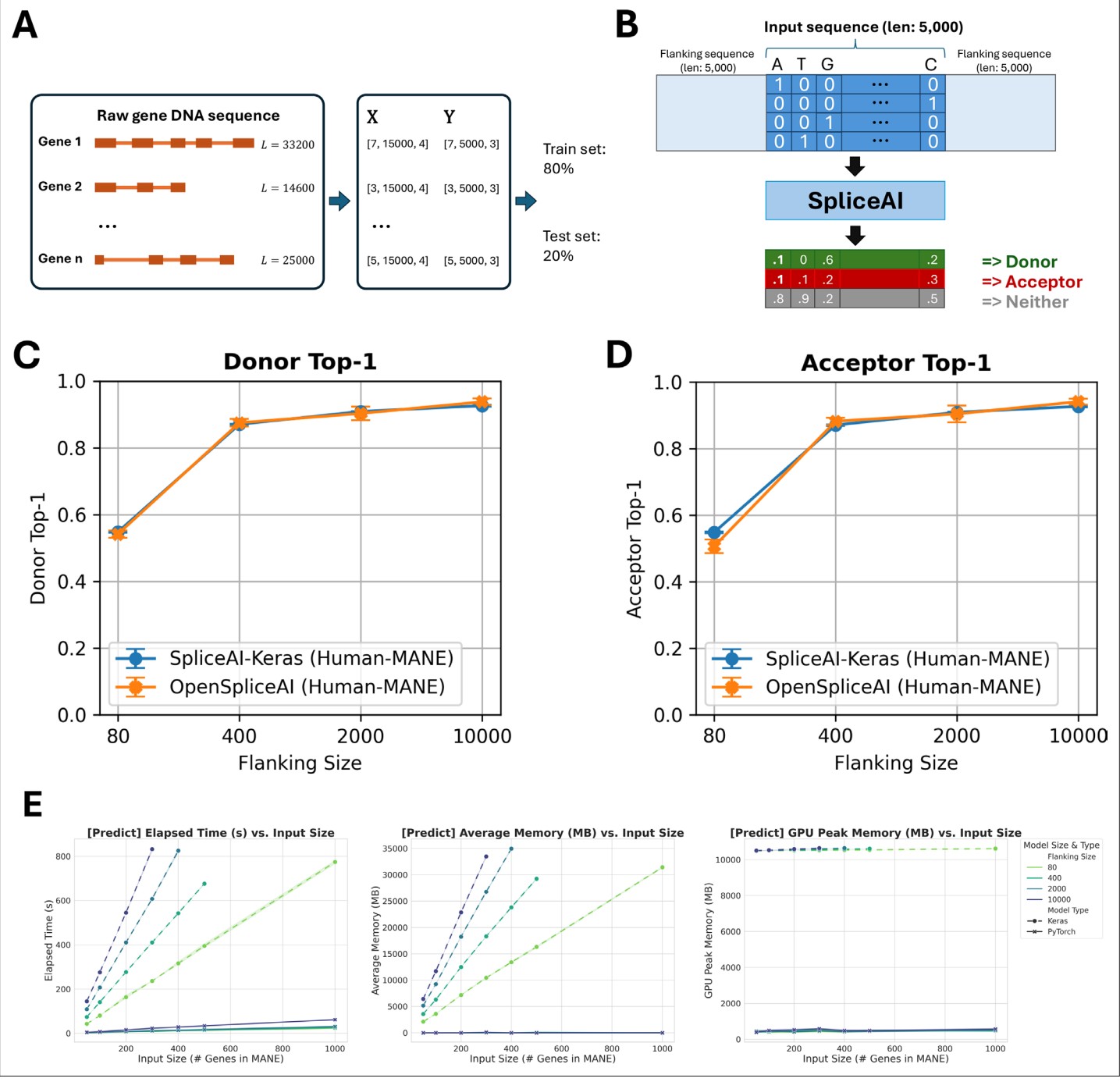

**Figure 2.** Overview of OpenSpliceAI framework, performance benchmarking, and comparison with SpliceAI-Keras. (**A**) Schematic overview of OpenSpliceAI's approach. Gene sequences are first extracted from the genome FASTA file and one-hot encoded (*X*). Splice sites are identified and labeled using the annotation file (*Y*). The resulting paired data (*X*, *Y*) for each gene is then compiled for model training (80% of the sequences) and testing (20% of the sequences). (**B**) Workflow of the OSAI$_{MANE}$ 10,000 model. Input sequences are one-hot encoded and padded with 5000 Ns ([0,0,0,0]) on each side, totaling 10,000 Ns. The model processes the input and outputs, for each position, the probability of that position being a donor site, an acceptor site, or neither. (**C–D**) Performance comparison between OSAI$_{MANE}$ and SpliceAI-Keras on splicing donor and acceptor sites, trained with 80 nt, 400 nt, 2000 nt, and 10,000 nt flanking sequences. Evaluation metrics include top-1 accuracy for both donor and acceptor sites. Blue curves represent SpliceAI-Keras, while orange curves represent OSAI$_{MANE}$. Each dot is the mean over five trained model variants, and error bars show ±1 standard error of the mean. Performance is compared across test datasets from humans. (**E**) Benchmarking results for elapsed time, average memory usage, and GPU peak memory for the prediction submodule.

The online version of this article includes the following figure supplement(s) for figure 2:

*Figure 2 continued on next page*

*Figure 2 continued*

**Figure supplement 1.** Comparison of splice site prediction performance between SpliceAI-Keras (blue) and OSAI$_{MANE}$ (orange) across human (*Homo sapiens*) datasets with varying flanking sequence lengths.

**Figure supplement 2.** Comparison of splice site prediction performance between SpliceAI-Keras (blue) and OSAI$_{MANE}$ (orange) across house mouse (*Mus musculus*) datasets with varying flanking sequence lengths.

**Figure supplement 3.** Comparison of splice site prediction performance between SpliceAI-Keras (blue) and OSAI$_{MANE}$ (orange) across honeybee (*Apis mellifera*) datasets with varying flanking sequence lengths.

**Figure supplement 4.** Comparison of splice site prediction performance between SpliceAI-Keras (blue) and OSAI$_{MANE}$ (orange) across zebrafish (*Danio rerio*) datasets with varying flanking sequence lengths.

**Figure supplement 5.** Comparison of splice site prediction performance between SpliceAI-Keras (blue) and OSAI$_{MANE}$ (orange) across *Arabidopsis thaliana* datasets with varying flanking sequence lengths.

**Figure supplement 6.** Comparison of runtime and memory metrics for 'predict' (panels A–F) and 'variant' (panels G–L) in OSAI$_{MANE}$ models with different flanking sequences.

a mammal (mouse, *M. musculus*), an insect (honeybee, *A. mellifera*), a freshwater fish (zebrafish, *D. rerio*), and a flowering plant (thale cress, *A. thaliana*). Using OpenSpliceAI with the same training hyperparameters as the human model, we trained species-specific models that we designated OSAI$_{Mouse}$, OSAI$_{Zebrafish}$, OSAI$_{Honeybee}$, and OSAI$_{Arabidopsis}$ (see *Table 1*).

Training and test sets for each species were generated using the 'create-data' submodule (Methods). Due to differences in genome sizes across species, we report the number of protein-coding genes used for training and testing in each model (*Figure 3A*), along with statistics on the ratio of canonical to noncanonical splice sites (*Figure 3—figure supplement 1A*) and intron length distributions (*Figure 3—figure supplement 1B*) for human MANE and the four other selected species. To ensure that the test sets did not contain paralogs of the training sets, OpenSpliceAI aligns them using minimap2 (*Li, 2018*) and excludes test sequences with over 80% similarity and coverage to the training sequences and enforces a 20% sequence diversity threshold (see Methods). In its original paper, SpliceAI was evaluated on a test set containing genes from human chromosomes 1, 3, 5, 7, and 9, which the Ensembl database (http://grch37.ensembl.org/biomart/martview) classifies as free of paralogs. However, applying our paralog removal criteria, we found that 0.71% of the MANE transcripts from these chromosomes were paralogous to training set sequences. In other species, the proportion of removed paralogous sequences was 3.86% for mouse, 31.97% for zebrafish, 0.08% for honeybee, and 2.26% for *Arabidopsis* (*Figure 3B*).

As we did with OSAI$_{MANE}$, we retrained each species model five times using different random seeds (10–14) and evaluated performance based on top-1 accuracy, F1 scores, precision, recall, and AUPRC for donor and acceptor sites. The results for OSAI$_{Mouse}$, OSAI$_{Zebrafish}$, OSAI$_{Honeybee}$, and OSAI$_{Arabidopsis}$ are shown in *Figure 3*, *Figure 3—figure supplements 2–5*. We calculated the average percentage improvement across all flanking sequence sizes for donor and acceptor sites under four flanking

**Table 1.** Genome assembly and annotation details for species used for OpenSpliceAI training and transfer learning in this study. Note: For each species, the table includes the GenBank accession number, assembly name, ftp sites for assembly and annotation downloads, and annotation release dates.

| Species | Name | Genbank accession | Download link | Annotation release date |
|---|---|---|---|---|
| *H. sapiens* | GRCh38.p14 | GCA_000001405.29 | https://ftp.ncbi.nlm.nih.gov/genomes/all/annotation_releases/9606/GCF_000001405.40-RS_2023_03/ | March 21, 2023 |
| *M. musculus* | GRCm39 | GCA_000001635.9 | https://ftp.ncbi.nlm.nih.gov/genomes/all/GCF/000/001/635/GCF_000001635.27_GRCm39/ | February 8, 2024 |
| *A. mellifera* | Amel_HAv3.1 | GCA_003254395.2 | https://ftp.ncbi.nlm.nih.gov/genomes/all/GCF/003/254/395/GCF_003254395.2_Amel_HAv3.1/ | September 30, 2022 |
| *A. thaliana* | TAIR10.1 | GCA_000001735.2 | https://ftp.ncbi.nlm.nih.gov/genomes/all/GCF/000/001/735/GCF_000001735.4_TAIR10.1/ | June 16, 2023 |
| *D. rerio* | GRCz11 | GCA_000002035.4 | https://ftp.ncbi.nlm.nih.gov/genomes/all/GCF/000/002/035/GCF_000002035.6_GRCz11/ | August 15, 2024 |

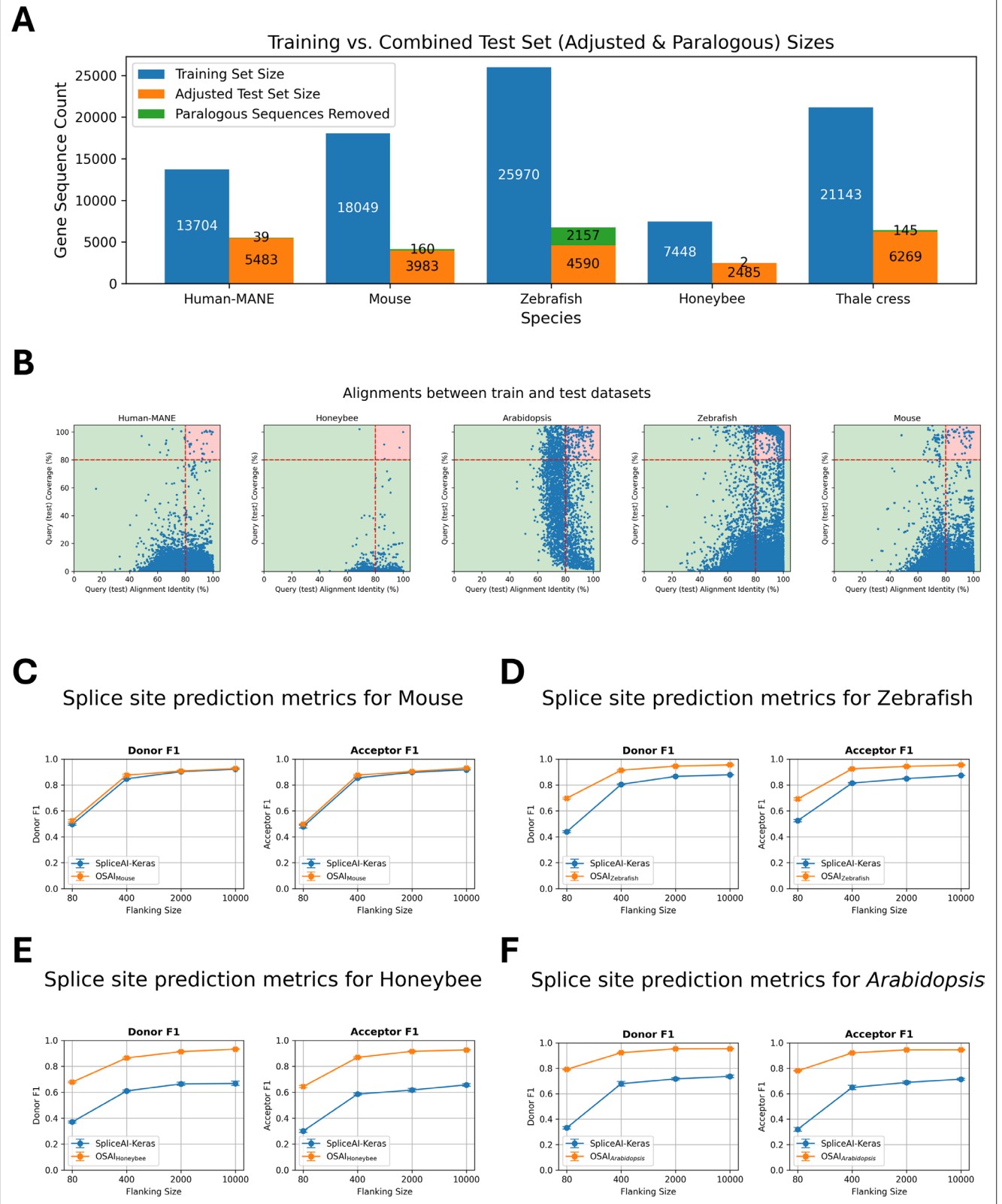

**Figure 3.** Cross-species dataset composition, sequence filtering, and performance comparison of OpenSpliceAI with SpliceAI-Keras. (**A**) The number of protein-coding genes in the training and test sets, along with the count of paralogous genes removed for each species: Human MANE, mouse, zebrafish, honeybee, and *Arabidopsis*. (**B**) Scatter plots of DNA sequence alignments between testing and training sets for human MANE, mouse, honeybee, zebrafish, and *Arabidopsis*. Each dot represents an alignment, with the *x*-axis showing alignment identity and the *y*-axis showing alignment

_Figure 3 continued_

coverage. Alignments exceeding 80% for both identity and coverage are highlighted in the red-shaded region and excluded from the test sets. (**C–F**) Performance comparisons of OSAIs trained on species-specific datasets (mouse, zebrafish, honeybee, and _Arabidopsis_) vs. SpliceAI-Keras, original published SpliceAI models, trained on human data. The orange curves represent OSAI metrics, while the blue curves show SpliceAI-Keras metrics. Each subplot (**C–F**) includes F1 score evaluated separately for donor and acceptor sites. Each data point represents the mean across five independently trained models, with error bars indicating the standard error.

The online version of this article includes the following figure supplement(s) for figure 3:

**Figure supplement 1.** Splice site motif count and intron length distributions across five species.

**Figure supplement 2.** Splice site prediction metrics for the mouse (_M. musculus_) across varying flanking sequence lengths.

**Figure supplement 3.** Splice site prediction metrics for the honeybee (_A. mellifera_) across varying flanking sequence lengths.

**Figure supplement 4.** Splice site prediction metrics for the zebrafish (_D. rerio_) across varying flanking sequence lengths.

**Figure supplement 5.** Splice site prediction metrics for the _A. thaliana_ across varying flanking sequence lengths.

sequence lengths. On average, OSAI outperformed SpliceAI-Keras by 2% in mouse, 54% in honeybee, 19% in zebrafish, and 57% in _Arabidopsis_.

The human and mouse genomes share a majority of their protein-coding genes (**_Waterston et al., 2002_**). This conserved evolutionary relationship likely explains the comparable performance of $OSAI_{Mouse}$ and SpliceAI-Keras. In contrast, OSAIs that had been retrained substantially outperformed SpliceAI-Keras in more distantly related species, particularly in honeybee, zebrafish, and _Arabidopsis_ (see Discussion).

## Adapting $OSAI_{MANE}$ to different species via transfer learning

Transfer learning can improve model performance by leveraging knowledge from related source domains (**_Zhuang et al., 2021_**). In the context of splice site prediction, we tested whether $OSAI_{MANE}$, initially trained on human splice annotations, could be effectively adapted to predict splice sites across other species.

We evaluated four species – _M. musculus_, _A. mellifera_, _D. rerio_, and _A. thaliana_ – by fine-tuning five distinct pretrained $OSAI_{MANE}$ models for each species. For every species, each pretrained model was fine-tuned using the same training and test datasets, yielding five transfer-trained variants. These variants – collectively referred to as $OSAI_{Mouse}$-transfer, $OSAI_{Honeybee}$-transfer, $OSAI_{Zebrafish}$-transfer, and $OSAI_{Arabidopsis}$-transfer – were directly compared with models trained from scratch ($OSAI_{Mouse}$, $OSAI_{Honeybee}$, $OSAI_{Zebrafish}$, and $OSAI_{Arabidopsis}$) to assess the benefits of transfer learning. For each species, we trained transfer-learned models using flanking sequences of 80 nt, 400 nt, 2000 nt, and 10,000 nt. We then compared the performance of transfer-trained and scratch-trained models by evaluating top-1 accuracy for donor and acceptor splice site predictions across four lengths of flanking sequences: 80, 400, 2000, and 10,000 nt (**_Figure 4A–D_**). Full results, including top-1 accuracy, F1 score, and AUPRC for donor and acceptor splice sites are provided in **_Figure 4—figure supplements 1–5_**.

Across all configurations, transfer-trained models consistently outperformed scratch-trained models in both accuracy and training stability, as evidenced by higher top-1 accuracies and lower standard errors across the five pretrained models. Notably, transfer-trained models achieved near-optimal performance after just one epoch, while scratch-trained models required 10 epochs to reach comparable results and showed substantial performance gaps between one and ten epochs (**_Figure 4E–H_**). Moreover, transfer learning also solved a convergence issue for one dataset: in the _A. thaliana_ scratch-training experiments using 10k flanking sequences, the CosineAnnealingWarmRestarts scheduler led to unstable optimization. Although switching to MultiStepLR with learning rate decay improved stability, reaching convergence was still a lengthy process. Notably, transfer learning did not display this problem (see the Methods section for training parameters).

After 10 epochs, transfer-trained models slightly outperformed their scratch-trained counterparts for _Arabidopsis_ and honeybee, the two species with the smallest genome sizes among those tested. These results suggest that pre-training improves generalization, particularly for compact genomes.

## Calibrating OpenSpliceAI models

Model calibration helps align predicted probabilities with the true likelihood of observed outcomes, thereby mitigating the risk of overconfident or underconfident predictions. Here, we applied class-wise

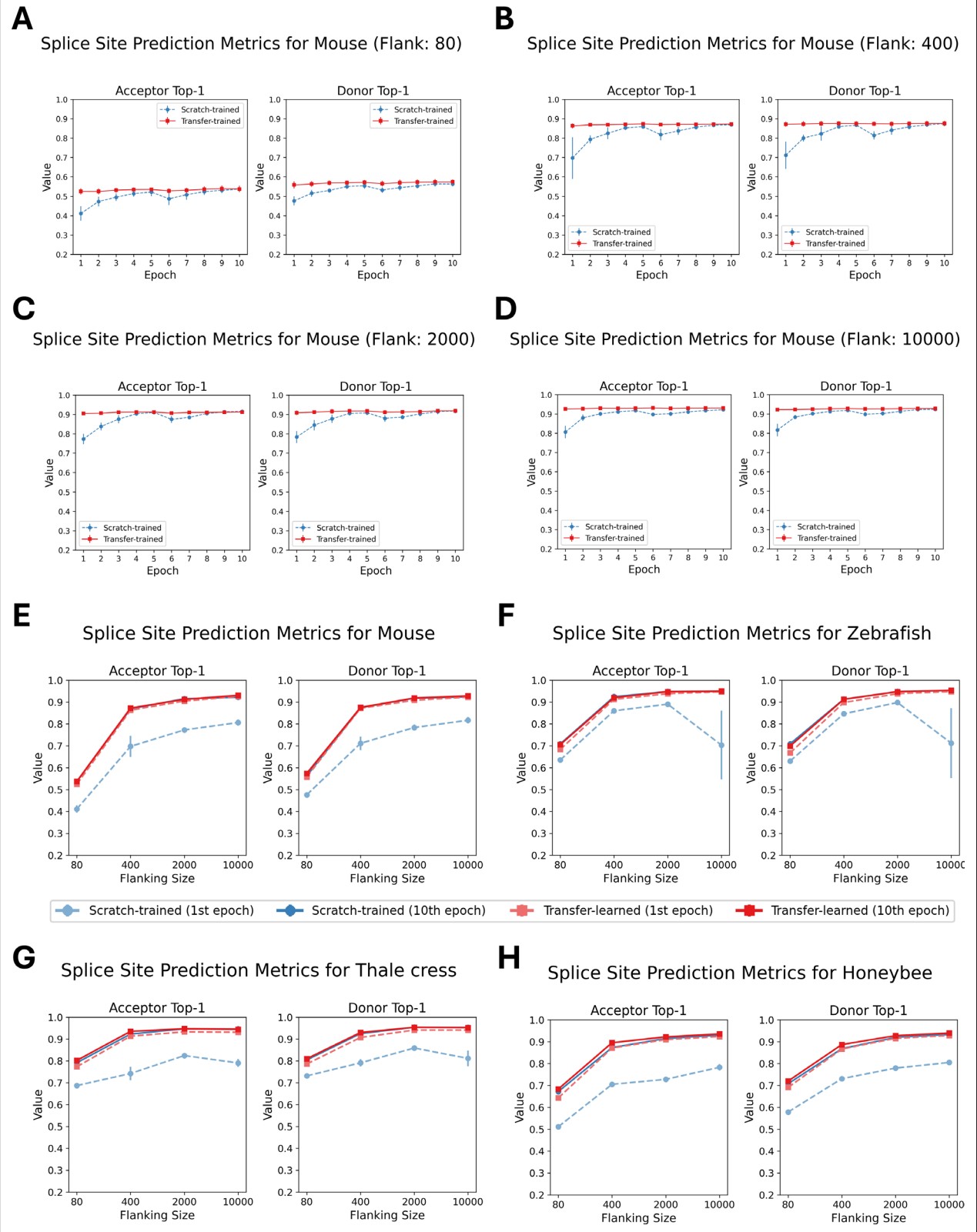

**Figure 4.** Performance comparison of scratch-trained and transfer-trained OSAIs across species and sequence lengths. (**A–D**) Top-1 accuracy for donor and acceptor splice sites of 80 nt, 400 nt, 2000 nt, and 10,000 nt models, comparing OSAI$_{Mouse}$ (scratch-trained) and OSAI$_{Mouse}$-transferred (transfer-trained) models over epochs 1–10 on the test dataset. (**E–H**) Top-1 accuracy after one epoch of training vs. after 10 epochs for both scratch-trained and transfer-trained models across the same sequence lengths. Each plot represents one species and its corresponding transfer-trained model: (**E**) OSAI$_{Mouse}$

*Figure 4 continued on next page*

*Figure 4 continued*

vs. OSAI$_{Mouse}$-transferred, (**F**) OSAI$_{Zebrafish}$ vs. OSAI$_{Zebrafish}$-transferred, (**G**) OSAI$_{Arabidopsis}$ vs. OSAI$_{Arabidopsis}$-transferred, and (**H**) OSAI$_{Honeybee}$ vs. OSAI$_{Honeybee}$-transferred. Each data point represents the mean across five independently trained models, with error bars indicating the standard error.

The online version of this article includes the following figure supplement(s) for figure 4:

**Figure supplement 1.** Transfer learning from OSAI$_{MANE}$ was leveraged to evaluate performance metrics for splice site prediction in mouse (*M. musculus*) across four flanking sequence lengths (80, 400, 2000, and 10,000 nt) and two splice site types (acceptor and donor).

**Figure supplement 2.** Transfer learning from OSAI$_{MANE}$ was leveraged to evaluate performance metrics for splice site prediction in honeybee (*A. mellifera*) across four flanking sequence lengths (80, 400, 2000, and 10,000 nt) and two splice site types (acceptor and donor).

**Figure supplement 3.** Transfer learning from OSAI$_{MANE}$ was leveraged to evaluate performance metrics for splice site prediction in zebrafish (*D. rerio*) across four flanking sequence lengths (80, 400, 2000, and 10,000 nt) and two splice site types (acceptor and donor).

**Figure supplement 4.** Transfer learning from OSAI$_{MANE}$ was leveraged to evaluate performance metrics for splice site prediction in *A. thaliana* across four flanking sequence lengths (80, 400, 2000, and 10,000 nt) and two splice site types (acceptor and donor).

**Figure supplement 5.** Cross-species transfer learning performance on human splice-site prediction.

temperature scaling, a single-parameter variant of Platt scaling, to adjust each class's predicted probabilities without altering the model's classification performance (see Methods). We calibrated OSAI$_{MANE}$ models on the validation set and subsequently evaluated them on the test set.

We then compared OSAI$_{MANE}$ models before and after calibration using reliability diagrams (***Figure 5A***, ***Figure 5—figure supplement 1***), which show reliability curves for non-splice, acceptor, and donor sites of OSAI$_{MANE}$ trained with flanking sequence lengths of 80, 400, 2000, and 10,000 nt, with the calibration temperature ($Ts$) in the legend. Calibration quality was quantified using negative log-likelihood (NLL) loss and expected calibration error (ECE). For each species, metrics were averaged over five calibrated models, and the results indicated slight improvements in both measures following calibration (***Figure 5B***, ***Figure 5—figure supplements 2 and 3***; see Methods). Temperature parameters greater than one indicated overconfidence, whereas values below one indicated underconfidence. After calibration, score distributions for donor and acceptor sites shifted slightly away from extreme values (1 and 0), resulting in smoother probability distributions (***Figure 5C***).

We observed similar outcomes when calibrating OSAIs on mouse (***Figure 5—figure supplement 4***), honeybee (***Figure 5—figure supplement 5***), zebrafish (***Figure 5—figure supplement 6***), and *Arabidopsis* (***Figure 5—figure supplement 7***). We observed very small changes after calibration across phylogenetically diverse species, suggesting that OpenSpliceAI's training regimen yielded well-calibrated models, although it is possible that a different calibration algorithm might produce further improvements in performance.

## Comparing OSAI$_{MANE}$ and SpliceAI via variant effects of ISM

A crucial finding of ***Jaganathan et al., 2019***, was that SpliceAI was capable of capturing nonlocal effects of genomic mutations on splice site location and strength. In order to show that OSAI$_{MANE}$ has the same capabilities, we recreated several of their studies, as well as a large-scale ISM experiment aimed at elucidating the model's learned splice site recognition pattern.

First, we recreated the experiment from Jaganathan et al. in which they mutated every base in a window around exon 9 of the U2SURP gene and calculated its impact on the predicted probability of the acceptor site. We repeated this experiment on exon 2 of the DST gene, again using both SpliceAI and OSAI$_{MANE}$. In both cases, we found a strong similarity between the resultant patterns between SpliceAI and OSAI$_{MANE}$, as shown in ***Figure 6A***. To evaluate concordance more broadly, we randomly selected 100 donor and 100 acceptor sites and performed the same ISM experiment on each site. The Pearson correlation between SpliceAI and OSAI$_{MANE}$ yielded an overall median correlation of 0.857 (see Methods; additional DNA logos in ***Figure 6—figure supplement 1***).

To characterize the local sequence features that both models focus on, we computed the average decrease in predicted splice site probability resulting from each of the three possible single-nucleotide substitutions at every position within 80 bp for 100 donor and 100 acceptor sites randomly sampled from the test set (chromosomes 1, 3, 5, 7, and 9). ***Figure 6B*** shows the average decrease in splice site strength for each mutation in the format of a DNA logo, for both tools. Pearson correlation analysis of the position weight matrices per base (***Gupta et al., 2007***) yielded a similarity of 0.996 for the donor and 0.997 for the acceptor between OSAI$_{MANE}$ and SpliceAI DNA logos, demonstrating high similarity

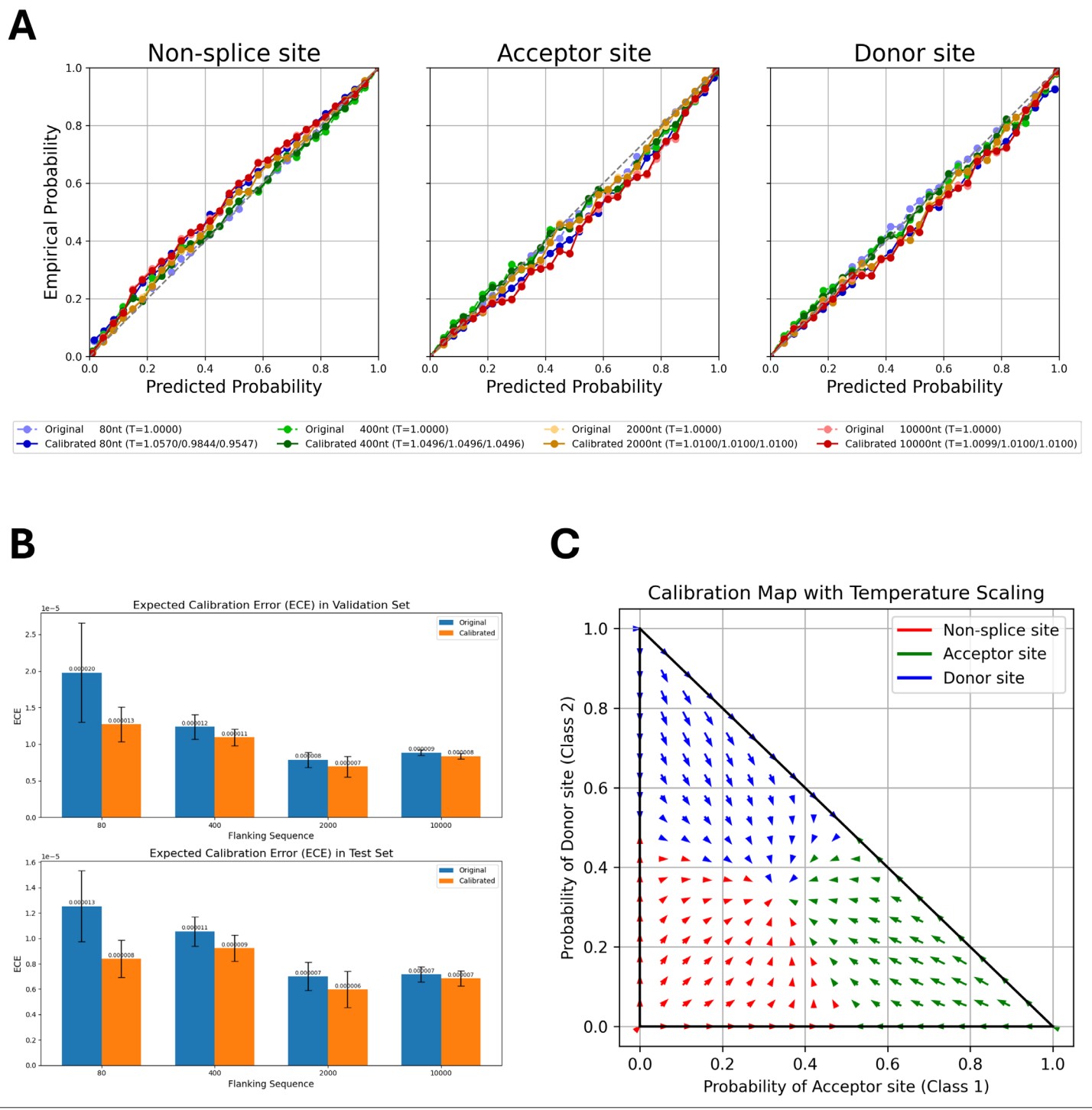

**Figure 5.** Calibration of OSAI$_{MANE}$ predictions across splice-site classes and flanking sequence lengths. (**A**) Calibration results for OSAI$_{MANE}$ across non-splice sites, acceptor sites, and donor sites. Models trained with different flanking sequence lengths are represented by color: 80 nt (blue), 400 nt (green), 2000 nt (orange), and 10,000 nt (red). Dotted curves in lighter colors denote pre-calibration results, while solid curves in darker shades show post-calibration results. (**B**) Expected calibration error (ECE) on the validation set (top) and test set (bottom), comparing the OSAI$_{MANE}$'s performance before (blue bars) and after (orange bars) calibration. For each flanking sequence OSAI$_{MANE}$, five calibration experiments were performed, with the mean loss and ±1 standard error. (**C**) Two-dimensional calibration map for OSAI$_{MANE}$, illustrating how raw predicted probabilities for acceptor (*x*-axis) and donor (*y*-axis) sites are transformed after calibration. Arrows indicate the shift from pre- to post-calibration states in two-dimensional probability space, resulting in a smoother probability distribution.

The online version of this article includes the following figure supplement(s) for figure 5:

*Figure 5 continued on next page*

*Figure 5 continued*

**Figure supplement 1.** Calibration results for human MANE splice site classification at four flanking sequence sizes.

**Figure supplement 2.** Expected calibration error (ECE) on the validation (top) and test (bottom) sets.

**Figure supplement 3.** Negative log likelihood (NLL) loss on the validation (top) and test (bottom) sets.

**Figure supplement 4.** Calibration results for house mouse (*M. musculus*) splice site classification at four flanking sequence sizes.

**Figure supplement 5.** Calibration results for zebrafish (*D. rerio*) splice site classification at four flanking sequence sizes.

**Figure supplement 6.** Calibration results for honeybee (*A. mellifera*) splice site classification at four flanking sequence sizes.

**Figure supplement 7.** Calibration results for *A. thaliana* splice site classification at four flanking sequence sizes.

between the two tools, with the strongest signals observed for mutations at the donor/acceptor sites outside the canonical GT/AG dinucleotide motif. The acceptor sites additionally show relatively higher sensitivity to A and G just upstream of the acceptor site, which is expected due to the CT richness of the polypyrimidine tracts common in this region (*Majewski and Ott, 2002*).

Jaganathan et al. also demonstrated SpliceAI's ability to predict cryptic splicing mutations – intronic mutations that create alternatively spliced transcripts. We recreated their experiment in which they investigated an MYBPC3 intron mutation associated with cardiomyopathy. Both $OSAI_{MANE}$ and SpliceAI predict very similar changes in the location and strength of acceptor site gain and loss events (*Figure 6C*). We then extended this experiment by examining an intronic splicing mutation in the OPA1 gene which has been shown to cause alternative splicing of a cryptic pseudoexon upstream (*Qian et al., 2021*). Again, both $OSAI_{MANE}$ and SpliceAI correctly predicted this event with similarly high accuracy.

We then replicated Jaganathan et al.'s experiment on the CFTR gene, in which they showed that SpliceAI predicted all of the splice sites accurately without any false positives. Using the full gene sequence from the GRCh38 assembly and a score threshold of 0.5, we found that $OSAI_{MANE}$ and SpliceAI predict the exact same set of donor and acceptor sites, and accurately capture all but the first donor site, using the MANE Select annotation as reference (*Figure 6D*).

## Discussion

We developed OpenSpliceAI to be a modular Python toolkit designed as an open-source implementation of SpliceAI, to which we added several key enhancements. The framework replicates the core logic of the SpliceAI model while optimizing prediction efficiency and variant effect analysis, such as acceptor and donor gains or losses, using pre-trained models. Our benchmarks show substantial computational advantages over SpliceAI, with faster processing, lower memory usage, and improved GPU efficiency (*Figure 2B*, *Figure 2—figure supplement 6*). These improvements are driven by our optimized PyTorch implementation that employs dynamic computation graphs and on-demand GPU memory allocation – allowing memory to be allocated and freed as needed – in contrast to SpliceAI's static, Keras-based TensorFlow approach, which pre-allocates memory for the worst-case input size. In SpliceAI, this rigid memory allocation leads to high memory overhead and frequent out-of-memory errors when handling large datasets through large loop iteration prediction. Additionally, OpenSpliceAI leverages streamlined data handling and enhanced parallelization through batch prediction and multiprocessing, automatically distributing tasks across available threads. Together, these features prevent the memory pitfalls common in SpliceAI and make OpenSpliceAI a more scalable and efficient solution for large-scale genomic analysis.

It is important to note that even though OpenSpliceAI and SpliceAI share the same model architecture, the released trained models are not identical. The variability observed between our models and the original SpliceAI – and even among successive training runs using the same code and data – can be attributed to several sources of inherent randomness. First, weight initialization is performed randomly for many layers, which means that different initial weights can lead to distinct convergence paths and final model parameters. Second, the process of data shuffling alters the composition of mini-batches during training, impacting both the training dynamics and the statistics computed in batch normalization layers. Although batch normalization is deterministic for a fixed mini-batch, its reliance on batch statistics introduces variability due to the random sampling of data. Finally, OpenSpliceAI employs the AdamW optimizer (*Loshchilov and Hutter, 2019*), which incorporates exponential moving averages

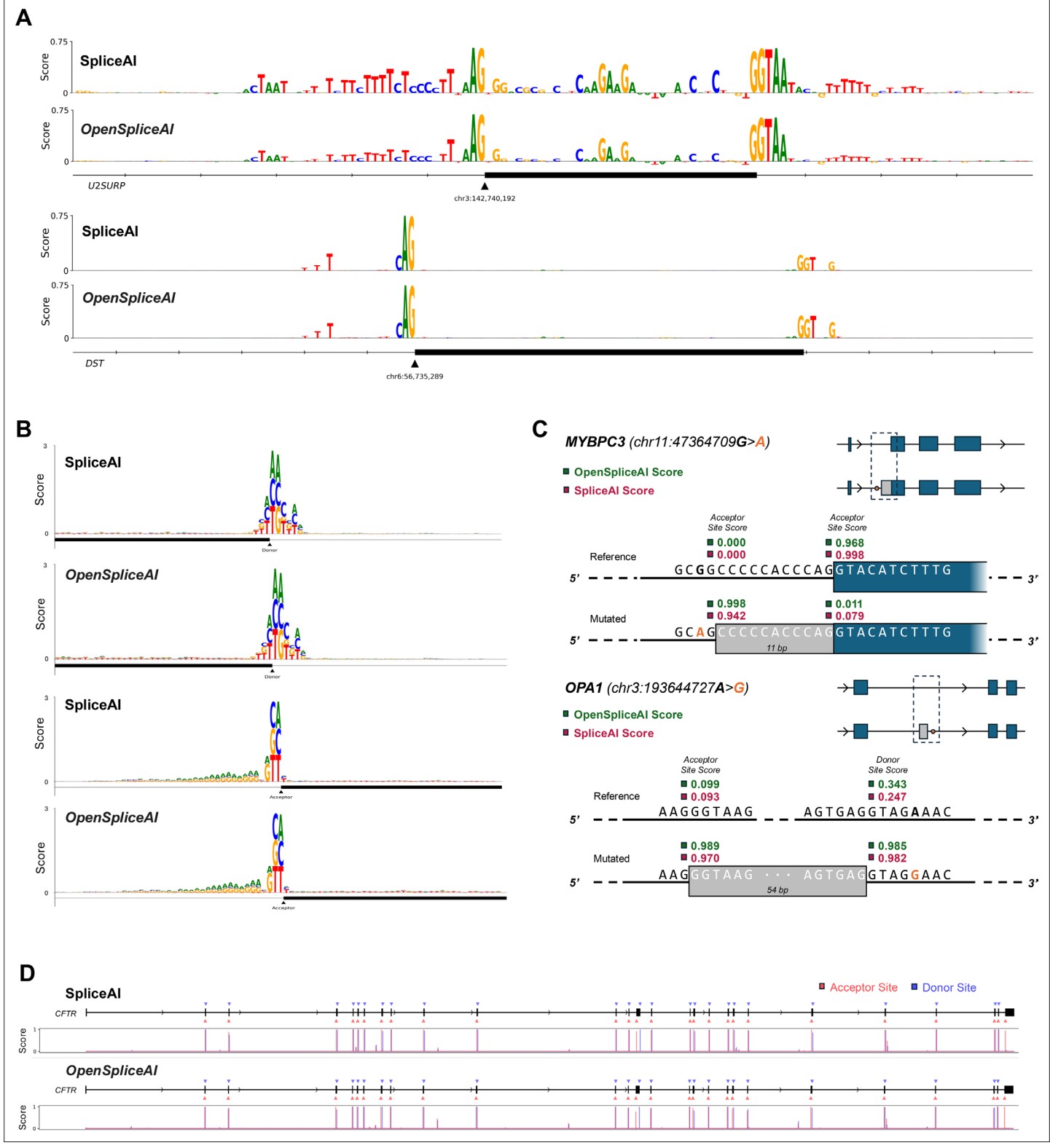

**Figure 6.** Comparison of SpliceAI and OSAI$_{MANE}$ in predicting mutation impacts and cryptic splicing events. (**A**) Plot of importance scores for nucleotides near the acceptor site of exon 9 of U2SURP (top) and DST (bottom), for both SpliceAI and OSAI$_{MANE}$. The importance score is calculated by taking the average decrease in acceptor site score across the three possible point mutations at a given base position. (**B**) Plot of the impact of each possible point mutation within 80 bp of a donor (top) site or acceptor (bottom) site, for both SpliceAI and OSAI$_{MANE}$. The impact is the raw decrease in predicted splice site score after mutating a given base to a different one. (**C**) Visualization of cryptic splicing variants being predicted for the MYBPC3 gene (top), with

*Figure 6 continued on next page*

*Figure 6 continued*

an acceptor site gain and loss event, from SpliceAI's original analysis, and the OPA1 gene (bottom), where a cryptic exon inclusion event was recently reported (*Qian et al., 2021*). (**D**) Predicted splice sites for the entire CFTR gene, with the corresponding predicted probability distribution by base position plotted below, for both SpliceAI and OSAI$_{MANE}$.

The online version of this article includes the following figure supplement(s) for figure 6:

**Figure supplement 1.** Zoomed-in 160 bp DNA sequence logos derived from in silico mutagenesis (ISM) importance score profiles for representative donor (**A–E**) and acceptor (**F–J**) splice sites.

of the first and second moments of the gradients. This mechanism serves a momentum-like role, contributing to an adaptive learning process that is inherently stochastic. Moreover, subtle differences in the order of operations or floating-point arithmetic, particularly in distributed computing environments, can further amplify this stochastic behavior. Together, these factors contribute to the observed nondeterministic behavior, resulting in slight discrepancies between our trained models and the original SpliceAI, as well as among successive training sessions under identical conditions.

OpenSpliceAI empowers researchers to adapt the framework to many other species by including modules that enable easy retraining. For closely related species such as mice, our retrained model demonstrated comparable or slightly better precision than the human-based SpliceAI model. For more distant species such as *A. thaliana*, whose genomic structure differs substantially from humans, retraining OpenSpliceAI yields much greater improvements in accuracy. Our initial release includes models trained on the human MANE genome annotation and four additional species: mouse, zebrafish, honeybee, and *A. thaliana*. We also evaluated pre-training on mouse (OSAI$_{Mouse}$), honeybee (OSAI$_{Honeybee}$), zebrafish (OSAI$_{Zebrafish}$), and *Arabidopsis* (OSAI$_{Arabidopsis}$) followed by fine-tuning on the human MANE dataset. While cross-species pre-training substantially accelerated convergence during fine-tuning, the final human splicing prediction accuracy was comparable to that of a model trained from scratch on human data. This result indicates that our architecture seems to capture all relevant splicing features from human training data alone and thus gains little or no benefit from cross-species transfer learning in this context (see *Figure 4—figure supplement 5*).

OpenSpliceAI also includes modules for transfer learning, allowing researchers to initialize models with weights learned on other species. In our transfer learning experiments, models transferred from human to other species displayed faster convergence and higher stability, with potential for increased accuracy. We also incorporate model calibration via temperature scaling, providing better alignment between predicted probabilities and empirical distributions.

The ISM study revealed that OSAI$_{MANE}$ and SpliceAI made predictions using very similar sets of motifs (*Figure 6B*). Across several experiments, we note that SpliceAI exhibits an inherent bias near the starts and ends of transcripts which are padded with flanking Ns (as was done in the original study), predicting donor and acceptor sites in these boundaries with an extremely high signal that disappears when the sequence is padded with the actual genomic sequence. For example, the model correctly predicted the first donor site of the CFTR gene when the gene's boundaries were flanked with N's; however, when replaced those N's with the actual DNA sequence upstream of the gene boundary, the signal all but disappeared, as seen in *Figure 6D*. This suggests a bias resulting from the way the model is trained. In our ISM benchmarks, we thus chose not to use flanking N's unless explicitly recreating a study from the original SpliceAI paper.

Additionally, we note that both the SpliceAI and OSAI$_{MANE}$ 'models' are the averaged result of five individual models, each initialized with slightly different weights. During the prediction process, each individual model was found to have discernibly different performance. By averaging their outputs leveraging the deep-ensemble approach (*Fort et al., 2019*; *Lakshminarayanan et al., 2017*), the overall performance of both SpliceAI and OpenSpliceAI improved while reducing sensitivity to local variations. In essence, this method normalizes the inherent randomness of the individual models, resulting in predictions that are more robust and better represent the expected behavior, ultimately yielding improved average performance across large datasets. OpenSpliceAI's 'predict' submodule averages across all five models by default, but it also supports prediction using a single model.

In summary, OpenSpliceAI is a fully open-source, accessible, and computationally efficient deep learning system for splice site prediction. Its modular architecture, enhanced performance, and adaptability to diverse species make it a powerful tool for advancing research on gene regulation and splicing across diverse species.

## Methods

OpenSpliceAI is designed with modular subcommands that allow users to preprocess genomic data into training and test sets, train models, perform model calibration, make efficient predictions, and conduct variant analysis. The following sections summarize the usage and technical implementation of each subcommand.

### OpenSpliceAI create-data subcommand

The 'create-data' subcommand converts standard genomics data formats into a machine-readable form suitable for training machine learning models. It processes genomic sequences (FASTA) and genome annotations (GFF/GTF) to produce gene sequences and splice site labels stored in Hierarchical Data Format version 5 (HDF5).

In this standard supervised sequence-to-sequence machine learning framework, a dataset comprising input features ($Xs$) and corresponding labels ($Ys$) is constructed. Here, $X$ represents the one-hot-encoded pre-mRNA sequences, which serve as the input variables for prediction, while $Y$ denotes the labels, specifically the donor and acceptor splice sites derived from the genome annotations.

For each gene locus, the longest transcript is selected as the canonical transcript, consistent with the canonical-transcript-labeling approach of SpliceAI. By default, the '--biotype' argument is set to 'protein-coding', which means only protein-coding genes are included in the feature set and label set. Users can change this setting to 'all' to include both protein-coding and non-coding genes.

#### Splitting gene sequences into training and testing sets

For generating datasets for OSAI$_{MANE}$, OpenSpliceAI adopts SpliceAI's chromosome-based partitioning strategy. In human datasets, the training set is defined by chromosomes 2, 4, 6, 8, 10, 11, 12, 13, 14, 15, 16, 17, 18, 19, 20, 21, 22, X, and Y, while the testing set is defined by chromosomes 1, 3, 5, 7, and 9. For nonhuman species, OpenSpliceAI defaults to a random splitting method (specified by the '--split-method' parameter). In this approach, the algorithm first computes the total chromosome length, randomly shuffles the chromosomes, and then iteratively assigns them to the training or testing set until the desired split ratio (defaulting to 80% training) is achieved, with any remaining chromosomes allocated to the test set.

#### Pseudogenes and paralogous gene sequences removal

To ensure the integrity and accuracy of model testing, pseudogenes – segments of DNA that resemble functional genes but are incapable of coding for a protein – are removed from the test dataset. This is accomplished by filtering out genes in the GFF file that either have 'pseudogene' as the feature type in the third column or specify 'pseudogene', 'transcribed_pseudogene', or 'processed_pseudogene' for the 'gene_biotype' or 'biotype' fields.

The removal of paralogous genes is also critical, as sequence similarity between training and test sets can lead to data leakage. OpenSpliceAI performs DNA sequence alignment to detect paralogous sequences. Specifically, OpenSpliceAI uses mappy, a Python wrapper for minimap2 (*Li, 2018*), to align test sequences to the training set, applying the '--asm20' argument to allow a sequence divergence of up to 20%. Following alignment, OpenSpliceAI examines each result and excludes any test sequence that shows more than 80% sequence similarity and 80% coverage compared to any sequence in the training set.

#### One-hot encoding scheme

The one-hot encoding procedure for the input sequence for model training, testing, and prediction uses the following representation: $A = [1, 0, 0, 0]$, $C = [0, 1, 0, 0]$, $G = [0, 0, 1, 0]$, $T (or\ U) = [0, 0, 0, 1]$. Any ambiguous nucleotide (denoted as N or other nonstandard symbols) is encoded as $[0, 0, 0, 0]$. The encoding of the labels for model training uses the scheme: none-splice site = $[1, 0, 0]$, acceptor site = $[0, 1, 0]$, donor site = $[0, 0, 1]$, padding = $[0, 0, 0]$. The predictions use the same scheme, where the three output channels sum to one, representing a probability score.

## Gene sequence segmentation for one-hot-encoded features (*X*s) and labels (*Y*s)

Following the approach used by SpliceAI, OpenSpliceAI divides gene sequences into overlapping segments, each spanning 15,000 nt. Each segment comprises a central region of 5000 nt, flanked on both sides by 5000 nt extensions, thereby providing essential upstream and downstream context. A step size of 5000 nt is used to ensure comprehensive coverage with overlapping windows. For instance, a gene that is 22,000 nt long is partitioned into five segments. Each segment is represented as a tensor with dimensions $(15,000, 4)$; when a segment lacks sufficient nucleotides, the remaining positions are padded with 'N' bases to maintain uniform tensor dimensions. Notably, the final segment may contain fewer real nucleotides – only 2000 in this example – with the deficit filled by padding. Thus, the one-hot-encoded feature matrix ($X$) for the gene has a shape of $(5, 15,000, 4)$, while the corresponding label tensor ($Y$), which focuses on the central 5000 nt, has a shape of $(5, 5,000, 3)$. The batch size is set to 100 by default, meaning that OpenSpliceAI concatenates the first dimension of each tensor from 100 genes into a single matrix and performs the same concatenation for the label tensor matrix.

## Selecting splice sites for inclusion in dataset labels (*Y*s)

Even with curated annotations, some splice sites in the annotation file may still be misannotated. To improve the accuracy of splice site labeling, OpenSpliceAI provides a '--canonical-only' argument that restricts analysis to canonical splice sites. By default, this option is disabled, so all splice sites in the annotation file are evaluated. These include the U2-snRNP-type motifs 'GT-AG' and 'GC-AG' (*Brow, 2002*) and the U12-snRNP-type motifs 'GT-AG' and 'AT-AC' (*Frilander and Steitz, 1999*; *Patel and Steitz, 2003*; *Wassarman and Steitz, 1992*).

## OpenSpliceAI train subcommand

After the training and test sets are created, this subcommand takes the HDF5 outputs from the 'create-data' subcommand and enables users to train their OpenSpliceAI model. Users can train different OpenSpliceAI models with various flanking sequence lengths, including 80 nt, 400 nt, 2000 nt, and 10,000 nt.

### OpenSpliceAI adaptive learning

OpenSpliceAI uses the AdamW optimizer (*Loshchilov and Hutter, 2019*) with a default learning rate of 0.001. The training dataset is further split into 90:10 for training and validation. By default, OpenSpliceAI trains a model for 10 epochs, with an early stopping patience of 2.

The '--scheduler' argument enables users to choose between two built-in PyTorch learning rate schedulers – MultiStepLR and CosineAnnealingWarmRestarts (*Loshchilov and Hutter, 2016*) – to dynamically adjust the learning rate during training. By default, OpenSpliceAI employs 'MultiStepLR' with a learning rate of 0.001, beginning with a 0.5 decay from the sixth epoch, the same approach used in the SpliceAI model.

As detailed in the Results section, training the OSAI model on *Arabidopsis* data using the Multi-StepLR scheduler resulted in a more stable training process. In contrast, all other OpenSpliceAI models, except OSAI$_{Arabidopsis}$, were trained using the 'CosineAnnealingWarmRestarts' scheduler, configured with 'T_0=5', 'T_mult = 1', 'eta_min = 1e-5', and 'last_epoch=-1'. This scheduler gradually reduces the learning rate from an initial value of 1e-3 to a minimum of 1e-5 in a smooth, wave-like (cosine) pattern over each cycle. The parameter 'T_0=5' sets the initial period for the cosine decay, meaning the learning rate completes one full cycle – from the starting rate down to 'eta_min' and back – within 5 epochs. After the first cycle, the learning rate 'restarts' at its initial value, creating a 'warm restart'.

### OpenSpliceAI loss function

By default, OpenSpliceAI uses the categorical cross-entropy loss function (*Equation 1*) to compute the loss at every nucleotide position in the input DNA sequence. This loss function measures the discrepancy between the predicted probability distribution and the true distribution for each position, which is standard practice for multi-class classification tasks. Alternatively, users can opt for the focal loss (*Equation 2*; *Lin et al., 2018*).

Focal loss enhances the standard cross-entropy loss by adding a modulating term, $(1 - P_{class})^{\gamma}$, where $P_{class}$ represents the model's predicted probability for the correct class. This term down-weights the loss assigned to well-classified examples, allowing the model to concentrate more on the misclassified or harder-to-classify cases. For instance, setting $\gamma$ to 2 amplifies the focus on challenging predictions, which is particularly beneficial in scenarios with class imbalance or when the signal in the data is subtle. This dynamic weighting of loss can enhance overall model accuracy.

$$Loss_{CEL} = \sum_{class \in \{donor, acceptor, neither\}} I_{class} \times \log\left(P_{class}\right) \tag{1}$$

$$Loss_{FL} = \sum_{class \in \{donor, acceptor, neither\}} I_{class} \times \left(1 - P_{class}\right)^{\gamma} \times log\left(P_{class}\right), where\ \gamma = 2 \tag{2}$$

## OpenSpliceAI transfer subcommand

Instead of training a model entirely from scratch, users can leverage transfer learning to adapt a human-trained model for a target species using the transfer subcommand. This process resembles standard model training but starts from a pre-trained model specified with the '--pretrained-model' argument. We recommend OSAI$_{MANE}$ as the pre-trained base model. Once the pre-trained weights are loaded, the transfer subcommand enables flexible fine-tuning. Users can either unfreeze all layers (using the '--unfreeze-all' flag) or selectively train the final layers (with '--unfreeze<INT >') to adapt the model more effectively to the target species data. In addition, similar to the train subcommand, the transfer subcommand integrates adaptive learning rate scheduling and early stopping to optimize convergence and prevent overfitting, all while using the same loss function configuration.

While transfer learning employs the same underlying OpenSpliceAI architecture, optimizer, scheduler, and loss function as training from scratch, it differs primarily in its initialization step, which is based on a fully trained model. For optimal results, we recommend selecting species with high-quality genome assemblies and comprehensive annotations, such as *H. sapiens*. This approach substantially reduces training time and can improve accuracy on the target species. See *Figure 4—figure supplements 1–4* for further details.

## OpenSpliceAI calibrate subcommand

One improvement in OpenSpliceAI over SpliceAI is the incorporation of model calibration, which refines model-predicted probabilities to align more closely with actual outcome likelihoods. This is achieved by calibrating the model's output so that a prediction with a probability of 0.6, e.g., accurately reflects a 60% chance of being correct. The calibrate subcommand evaluates scores around this value, identifies deviations from expected probabilities, and applies nonlinear adjustments to correct the score distribution without altering the model's performance. After the OpenSpliceAI model was trained, we used the validation dataset to calibrate the model and evaluated the calibrated results using the test dataset. Such calibration is crucial in predictive modeling, particularly for classification, as it ensures that predicted probabilities are consistent with observed outcomes. Uncalibrated models can be overconfident or underconfident, potentially compromising decision-making quality.

There are various methods for calibrating models, including Platt scaling (*Platt, 1999*), isotonic regression (*Zadrozny and Elkan, 2002*), and histogram binning (*Zadrozny and Elkan, 2001*). Here, we implemented class-wise temperature scaling, a variant of Platt scaling often used in knowledge distillation and statistical mechanics (*Hinton et al., 2015*; *Jaynes, 1957*). Temperature scaling is a post hoc adjustment that modifies model output probabilities to more accurately reflect true class likelihoods.

### OpenSpliceAI calibration optimization procedure

For model calibration in OpenSpliceAI, we froze the trained OpenSpliceAI model weights and augmented the network with a class-specific temperature scaling layer. Instead of using a single scalar temperature parameter, we employ a vector of temperature parameters $T = [T_0, T_1, T_2]$ corresponding to the non-splice site, acceptor site, and donor site, respectively. This design allows each class's logit to be scaled individually, thereby addressing the inherent class imbalance and the sparsity of splice site signals. The logits are divided by their corresponding temperature parameters before applying

the softmax function (**Bridle, 1989**), thereby aligning the predicted probabilities with the empirical likelihoods.

The temperature vector $T$ was optimized using the Adam optimizer (**Kingma and Ba, 2014**) with an initial learning rate of 0.01 for its adaptive capabilities. To further enhance convergence, we employ PyTorch's ReduceLROnPlateau scheduler, which reduces the learning rate by a factor of 0.1 if the validation loss does not improve over two consecutive epochs. In addition, early stopping was implemented with a patience of two epochs and a minimum improvement threshold (delta) of $10^{-6}$. If the validation loss did not decrease by at least $10^{-6}$ over two epochs, optimization halts early, ensuring calibration efficiency and preventing overfitting.

Temperature scaling modifies the logits $z$ (the raw outputs of the model before the softmax function) by scaling them with $T$ (**Equation 3**). The adjusted logits $z'$ are computed as:

$$z' = \frac{z}{T} \tag{3}$$

where $z$ represents the original logits. The calibrated probabilities $p$ are then obtained by applying the softmax function (**Equation 4**):

$$\hat{p} = softmax(z') \tag{4}$$

A higher temperature $T > 1$ spreads out the probability distribution, reducing confidence, while a lower temperature $T < 1$ sharpens it, increasing confidence. To optimize calibration, we use the negative log-likelihood (NLL) loss function defined as (**Equation 5**):

$$\boldsymbol{L_{NLL}} = -\sum_{i=1}^{N}\sum_{c=1}^{C} I_{i,c}\log\left(\hat{p}_{i,c}\right) \tag{5}$$

where $N$ is the number of samples, $C$ is the number of classes, in our case, acceptor site, donor site, and non-splice site. $I_{i,c}$ is the indicator function, which equals 1 if sample $i$ belongs to class $c$ and 0 otherwise. $\hat{p}_{i,c}$ is the model's predicted probability of the sample $i$ belongs to class $c$. A lower NLL indicates that the predicted probabilities are more closely aligned with the true labels, reflecting better calibration and overall model performance.

The optimal temperature $T^*$ is determined by minimizing the NLL loss over the validation dataset (**Equation 6**):

$$T^* = \arg\min_T \mathcal{L}_{NLL}(\hat{p}, y) \tag{6}$$

where $y$ are the true labels.

The temperature parameter $T$ was initialized to one and constrained between 0.05 and 5.0 to prevent extreme scaling. We employed gradient-based optimization to minimize the cross-entropy loss (nn.CrossEntropyLoss, i.e., NLL loss) on the validation set while keeping the original model weights fixed.

## OpenSpliceAI calibration evaluation

The first metric, which we used for temperature optimization, is NLL (**Equation 5**), which measures the match between predicted probabilities and the true labels; lower NLL values indicate better calibration.

We also evaluated the temperature-scaled new probabilities using the expected calibration error (ECE) (**Equation 7**). ECE quantifies the discrepancy between confidence estimates and actual accuracy over a range of probability bins. It does so by partitioning the predictions into $M$ bins, by default OpenSpliceAI sets to 30, and computing the weighted average of the absolute differences between the confidence (predicted probability) and accuracy within each bin. The ECE is defined as:

$$\mathcal{L}_{ECE} = \sum_{m=1}^{M} \frac{|B_m|}{N}\left|acc(B_m) - conf(B_m)\right| \tag{7}$$

$B_m$ is the set of indices of samples whose predicted probabilities fall into the $m$th bin; $|B_m|$ is the number of samples in the $m$th bin; $N$ is the total number of samples; $acc\left(B_m\right)$ is the average accuracy in the $m$th bin (**Equation 8**).

$$acc(B_m) = \frac{1}{|B_m|} \sum_{i \in B_m} \mathbf{1}(\hat{y}_i = y_i) \tag{8}$$

$conf(B_m)$ is the average confidence in the $m$th bin (**Equation 9**), where $\hat{y}_i$ is the predicted class for sample $i$, where $\hat{p}_i$ is the predicted probability associated with the predicted class $\hat{y}_i$.

$$conf(B_m) = \frac{1}{|B_m|} \sum_{i \in B_m} \hat{p}_i \tag{9}$$

An ECE of 0 indicates perfect calibration, where confidence and accuracy are aligned across all bins.

## OpenSpliceAI reliability curve and confidence interval

We generated calibration curves for each class using the calibration_curve function from scikit-learn, employing 30 bins with a uniform binning strategy. For input, the logits and labels were reshaped into 2D tensors. This function calculates the mean predicted probability and the true frequency of the positive class within each bin. To visualize the uncertainty in these estimates, we computed confidence intervals for each bin using the normal approximation method (**Raschka, 2018**; **Equations 10, 11, and 12**):

$$SE = \sqrt{\frac{\hat{p}(1 - \hat{p})}{n}} \tag{10}$$

$$CI_{lower} = \max(\hat{p} - z \cdot SE, 0) \tag{11}$$

$$CI_{upper} = \min(\hat{p} + z \cdot SE, 1) \tag{12}$$

where $p$ is the empirical probability and $n$ is the number of samples in the bin. **Equation 10** defines the standard error (SE) of the estimated parameter. For a 95% confidence level, $z$ is set to 1.96. **Equations 11 and 12** provide the lower and upper bounds of the confidence interval, respectively (see **Figure 5—figure supplements 1–7** for all reliability curve results).

## OpenSpliceAI predict subcommand

After the OpenSpliceAI model is trained, users can execute this subcommand to predict splice sites in DNA sequences provided in FASTA format. This command also supports limiting predictions to protein-coding genes by using a GFF annotation file for the given genome. It outputs the results in BED format, collecting all probable donor and acceptor site locations into separate files.

## OpenSpliceAI data preprocessing

Depending on the inputs of the subcommand, OpenSpliceAI will extract the input sequences differently. If only a FASTA file is provided, OpenSpliceAI will collect all sequences within the file for prediction. If a FASTA and GFF file are both provided, OpenSpliceAI will extract all features of type 'gene' from the GFF file and use those coordinates to extract sequences from the FASTA file for prediction.

To aid in memory management, OpenSpliceAI splits any sequence with length greater than the 'split-threshold' (default: 1,500,000 bases) into chunks that are no longer than this threshold. This parameter can be adjusted and ensures that each chunk can be loaded entirely into memory during the one-hot encoding process. Additionally, to optimize speed, if the total length of sequences in the FASTA file is below the 'hdf_threshold' (default: 5000 bases), the tool bypasses HDF5 compression and processes the input directly as text, achieving a slight performance speedup.

After all sequences are collected, the tool preprocesses the inputs using a method similar to that employed during training, but without handling any true labels. Each sequence is split into overlapping windows of size 5000 + 'flanking_size', where the overlap is equal to half of the 'flanking_size'. No clipping is allowed for the input, so if the final subsequence is shorter than 5000 + 'flanking_size', it is right-padded with N's. Similarly, the first subsequence is left-padded with 'flanking_size'/2 N's. If a sequence is split according to the 'split_threshold', it is divided so that adjacent subsequences share an overlap equal to half the 'flanking_size', preventing N-padding from interfering with predictions. This ensures that the model predicts every single base of the provided input sequences.

After window-based splitting, all 5000 + 'flanking_size' subsequences generated from the input FASTA entry are grouped together and one-hot encoded in parallel, yielding an entry with dimensions (N, 5000 + 'flanking_size', 4), where N is the number of subsequences generated from the given FASTA entry. These entries are further grouped into chunks of size 'chunk_size' (default 100), which ensures that they are processed together, reshaping the input to size (<='chunk_size', N, 5000 + 'flanking_size', 4). The resultant chunks are saved together as an HDF5-compressed file.

## OpenSpliceAI prediction algorithm design

To dynamically manage memory and optimize speed, OpenSpliceAI offers two modes of prediction: standard and turbo, which is controlled by the 'predict_all' flag. In standard mode, predictions for all bases are stored (which can be memory-intensive) and written to a BED file. In contrast, turbo mode skips storing individual predictions and directly converts them to a BED file without storing them, reducing memory usage. By default, turbo mode is enabled (see *Figure 1—figure supplement 2*).

To start the prediction process, OpenSpliceAI first loads the appropriate pre-trained models. Like SpliceAI, each pre-trained OpenSpliceAI model can include multiple individual models that are averaged to produce a final, higher-quality prediction. Users can specify either a single model file or a directory containing multiple models, in which case the tool automatically averages predictions from all provided models. Depending on the user's system, OpenSpliceAI selects the best available computing device for model loading and prediction, prioritizing CUDA, MPS, CPU in that order.

OpenSpliceAI performs batch prediction with parallelized prediction, significantly reducing prediction time. The 'batch_size' parameter is determined based on the 'flanking_size' and the computing device used. Chunked sequences are loaded into a PyTorch DataLoader object, which batches the one-hot-encoded chunk into dimension (<='batch_size', 4, 5000 + 'flanking_size'). Each batch is processed through all provided models, producing averaged, batched predictions, which are then accumulated by chunk and flattened to reconstruct the full input sequence (FASTA entry) of dimension (<='split_threshold', 3). The second dimension represents predictions for whether each base position is a donor site, acceptor site, or neither.

OpenSpliceAI's memory handling varies based on the prediction mode. In standard mode, the output tensor is periodically flushed to a file, controlled by the 'flush_predict_threshold' parameter, which specifies how many sequences are stored in memory before flushing to an HDF5 file. Predictions are then converted into a BED file that explicitly identifies donor and acceptor sites. In turbo mode, the prediction and BED-file-writing steps are performed simultaneously, and the raw tensor predictions for each base are discarded from memory rather than saved. This substantially reduces memory and processing time.

## OpenSpliceAI prediction outputs

The prediction step generates two BED files – one for donor sites and one for acceptor sites – containing the coordinates and scores of all splice sites that exceed a specified score threshold (set by the 'threshold' parameter, default 0.5). The tool automatically extracts relevant information from input files to determine splice site coordinates in the BED file. If not enough annotation data is provided (e.g. the FASTA header does not have transcript start and end coordinates and no annotation file is provided), the coordinates are reported relative to the FASTA sequence, with position 0 corresponding to the first nucleotide. If an annotation file is provided, the tool extracts protein-coding gene loci and calculates the coordinates of splice sites within each locus.

## OpenSpliceAI variant subcommand

The 'variant' subcommand reimplements SpliceAI's publicly available utility (*Jaganathan et al., 2019*) to evaluate the effects of genetic variants on the location and strength of splice sites. It does so by comparing predictions made on wild-type and mutant sequences to determine the impact of single nucleotide polymorphisms and insertions or deletions (INDELs) on the resulting mRNA transcript. The tool outputs 'delta' scores for four events: donor site gain, donor site loss, acceptor site gain, and acceptor site loss, along with the most probable position of each event relative to the mutation. It accepts a variant call format (VCF) file as input and returns an output VCF file annotated with the delta scores and positions.

This subcommand supports variant effect prediction using both PyTorch and Keras models to maintain compatibility with upstream workflows. However, PyTorch-based models are strongly recommended for faster prediction and lower memory overhead.

## OpenSpliceAI variant delta score calculation

The 'delta' score is defined similarly to *Jaganathan et al., 2019*, and refers to the maximum change in splicing score within a fixed window on each side of the mutation. By default, the window size is 50, meaning it will consider the donor and acceptor scores for the 101 positions around the variant. Supposing the array of donor and acceptor scores of the wild-type sequence are $d_{ref}$ and $a_{ref}$, and those of the mutated sequence are $d_{alt}$ and $a_{alt}$, then the delta scores (DS) are calculated as follows (*Equations 13–16*):

$$DS\left(Acceptor\,Gain\right) = \max\left(a_{alt} - a_{ref}\right) \tag{13}$$

$$DS\left(Acceptor\,Loss\right) = \max\left(a_{ref} - a_{alt}\right) \tag{14}$$

$$DS\left(Donor\,Gain\right) = \max\left(d_{alt} - d_{ref}\right) \tag{15}$$

$$DS\left(Donor\,Gain\right) = \max\left(d_{alt} - d_{ref}\right) \tag{16}$$

In *Jaganathan et al., 2019*, the term 'delta score' specifically refers to the maximum value among the four events. However, we do not use this score in our output. Instead, the output VCF file reports separate scores for each of the four events.

## OpenSpliceAI splice site variant scoring process

The 'variant' subcommand takes in a VCF file, a reference genome in FASTA format, the model, the flanking size, and a custom annotation file. It annotates each variant in the provided VCF with four delta scores and four corresponding 'delta positions', which represent the relative nucleotide location of each delta score. By default, the delta position is ±50, with negative values indicating positions upstream of the variant and positive values downstream. Variants outside of genes defined by the annotation file, those that are less than 'flanking_size' from the ends of the chromosome, and deletions greater than 2x 'distance' are excluded from annotation. The tool returns a VCF file with OpenSpliceAI annotations for all valid variants (*Figure 1—figure supplement 2*).

## OSAI$_{MANE}$ training

We generated training and test datasets using the 'create-data' subcommand. Following the SpliceAI approach, chromosomes 1, 3, 5, 7, and 9 were held out for testing. Paralogous genes were stringently removed from the test dataset based on sequence alignment results between the training and test sets (see Methods: Pseudogenes and paralogous gene sequences removal). The '--canonical-only' argument was used to label only donor and acceptor sites in U2-snRNP-type and U12-snRNP-type introns.

Next, we trained OSAI$_{MANE}$ with the train subcommand, employing a cosine annealing scheduler ('--scheduler CosineAnnealingWarmRestarts') and a categorical cross-entropy loss function ('--loss cross_entropy_loss') over 10 epochs ('--epochs 10').

We trained OSAI$_{MANE}$ with four different flanking sequence lengths: 80, 400, 2000, and 10,000 nt. For each flanking sequence length, five models were trained with different random seeds to enable ensemble score predictions, following the SpliceAI-Keras approach. Model performance was evaluated on the held-out chromosomes (1, 3, 5, 7, and 9) using top-k accuracy, AUPRC, overall accuracy, precision, recall, and F1 score for both donor and acceptor sites.

## Commands to train OSAI$_{MANE}$

The following commands reproduce the creation of OSAI$_{MANE}$ using the OpenSpliceAI framework. To do so, an annotation GFF file, specifically the Human RefSeq MANE v1.3 annotation, and a genome FASTA file, the GRCh38.p14 genome , are required.

- Creating training and test dataset:

```
openspliceai    create-data    --genome-fasta    GCF_000001405.40_GRCh38.
p14_genomic.fna    --annotation-gff    MANE.GRCh38.v1.3.refseq_genomic.gff
--output-dir train_test_dataset_MANE_test/ --remove-paralogs --min-identity
0.8 --min-coverage 0.8 --parse-type canonical --write-fasta --split-method
human --canonical-only
```

- Training OSAI$_{MANE}$

```
openspliceai train --flanking-size 10000 --train-dataset dataset_train.h5
--test-dataset dataset_test.h5 --output-dir model_train_outdir/ --project-
name OSAI-MANE --loss cross_entropy_loss --scheduler CosineAnnealingWarmRe-
starts --epochs 10
```

- Calibrating OSAI$_{MANE}$ (optional)

```
openspliceai calibrate –flanking-size 10000 --train-dataset dataset_train.
h5  --test-dataset  dataset_test.h5  --output-dir  model_calibrate_outdir/
--project-name OSAI-MANE-calibrate --pre-trained-model model_best.pt --loss
cross_entropy_loss
```

Similar to training OSAI$_{MANE}$, the OpenSpliceAI framework can also be used to train species-specific models using different genomes and genome annotations.

## Hardware resources for training in this study

OpenSpliceAI study was conducted on the Rockfish cluster. For data preprocessing, OpenSpliceAI was run with a single thread on a 24-core Intel Xeon Cascade Lake 6248R processor, with a base frequency of 3.0 GHz and a 1 TB NVMe local drive. The five OpenSpliceAI models – OSAI$_{MANE}$, OSAI$_{Mouse}$, OSAI$_{Zebrafish}$, OSAI$_{Honeybee}$, and OSAI$_{Arabidopsis}$ – were each trained with a single Nvidia A100 GPU with 40 GB of memory and 192 GB of DDR4 memory. Slurm jobs were submitted with '--mem=64 G'.

## Model architecture and training hyperparameters

Building on SpliceAI's model architecture, we re-implemented the deep residual CNN using PyTorch to improve flexibility and extensibility (see *Figure 1—figure supplement 1*; https://github.com/Kuanhao-Chao/OpenSpliceAI/blob/main/openspliceai/openspliceai.py). The network processes an input tensor of shape (batch size, input length, 4) that encodes one-hot nucleotide sequences. Four different flanking sequence lengths – 80, 400, 2000, and 10,000 nucleotides – are used to train four separate models, providing flexibility in capturing splicing signals at varying genomic contexts.

At the core of OpenSpliceAI, an initial 1D convolution (mapping from 4 channels to $L$ channels) projects the nucleotide embedding into a higher-dimensional feature space. The resulting features pass through a series of ResidualUnit blocks, each consisting of two dilated convolutional layers with LeakyReLU ($\alpha = 0.1$) activation and batch normalization. These dilated convolutional layers employ

**Table 2.** Summary of the four OpenSpliceAI model architectures, each trained with a distinct flanking sequence length (80, 400, 2000, and 10,000 nucleotides).

The table lists the kernel sizes (W), dilation rates (AR), number of residual and skip blocks, and total cropping length (CL).

| Parameter | Flanking = 80 | Flanking = 400 | Flanking = 2000 | Flanking = **10,000** |
|---|---|---|---|---|
| Kernel sizes (W) | [11, 11, 11, 11] | [11, 11, 11, 11, 11, 11, 11, 11] | [11, 11, 11, 11, 11, 11, 11, 11, 21, 21, 21, 21] | [11, 11, 11, 11, 11, 11, 11, 11, 21, 21, 21, 21, 41, 41, 41, 41] |
| Dilated rates (AR) | [1, 1, 1, 1] | [1, 1, 1, 1, 4, 4,,4,4] | [1, 1, 1, 1, 4, 4,,4,4, 10, 10, 10, 10] | [1, 1, 1, 1, 4, 4,,4,4, 10, 10, 10, 10, 25, 25, 25, 25] |
| Residual blocks | 4 | 8 | 12 | 16 |
| Skip connections | 1 (inserted after residual block 4) | 2 (inserted after residual blocks 4 and 8) | 3 (inserted after residual blocks 4, 8, and 12) | 4 (inserted after residual blocks 4, 8, 12, and 16) |
| Cropping length (CL); $\left(2 \times \sum \left[AR \times (W-1)\right]\right)$ | 80 | 400 | 2000 | 10,000 |

increasing dilation rates ($AR$ vector) and kernel sizes ($W$ vector) to enlarge the receptive field without requiring extremely deep networks. Every fourth residual block is followed by a Skip layer that merges skip-connection features via a 1D convolution, ensuring better gradient flow and stabilizing training.

To accommodate the shrinking of the sequence length necessitated by large receptive fields, we employ a Cropping1D layer that removes extra padding introduced by convolutional dilation. Specifically, the amount of cropping $CL$ is computed as twice the sum of $AR \times (W - 1)$. By slicing out $CL/2$ nucleotides from each end of the sequence, the Cropping1D layer aligns the network output with the desired prediction length. The final layer is a 1D convolution mapping the output features into three channels – representing probabilities for the donor site, acceptor site, or neither. We apply a softmax activation over these three positions at each nucleotide, yielding position-wise splice site prediction probabilities. See *Table 2* for a summary of the model architectures trained with four different flanking sequence lengths.

While model training is performed with fixed-length sequences (e.g. 15,000 nucleotides to produce 5000 prediction positions), the dilated convolutional structure and final Cropping1D layer allow each trained model to process variable-length sequences at inference time. Specifically, any sequence longer than the network's receptive field can be fed to the network, which will output predictions aligned to the valid region (i.e. the input length minus the cropping region). For flanking regions, users can either pad with N's or include relevant upstream and downstream genomic context. Memory permitting, this design grants flexibility in analyzing genomic segments of varying lengths in a single pass, without the need to retrain separate models for each new sequence length.

In total, these architectural components allow OpenSpliceAI to capture sequence contexts spanning up to 10,000 nucleotides. By varying kernel sizes, dilation rates, and cropping, the model can learn both local and long-range patterns important for accurate identification of canonical splice signals.

For model training, we introduced enhancements to the adaptive learning rate schedule. Specifically, we set a maximum learning rate for the initial 10 epochs and implemented an early stopping criterion to prevent overfitting. The adaptive learning rate decreases by a factor of 0.5 starting from the sixth epoch in the original implementation. Additionally, we incorporated the CosineAnnealingWarmRestarts scheduler to enable periodic learning rate restarts, which can help escape local minima and improve convergence.

## Evaluation metrics on model performance

To evaluate both scratch-trained and transfer-trained models, we used outputs from the 'create-data' subcommand derived from a test set that included only protein-coding genes and excluded paralogous sequences. For consistency with the SpliceAI study, we held out the same chromosomes for testing. For other species, we reserved approximately 20% of the data for testing and used the remaining 80% for training. These datasets were then used to assess (1) the original SpliceAI models, (2) all OSAI models trained from scratch, and (3) all OSAI models fine-tuned from OSAI$_{\text{MANE}}$.

### Top-k accuracy

In line with the definition used in the SpliceAI paper, we evaluate the model's performance using the top-k accuracy metric. This metric is computed by examining each dimension of the model's predictions. For each DNA sequence of length $L$, the model outputs an $L \times 3$ matrix: the first channel indicates non-splice sites, the second indicates acceptor splice sites, and the third indicates donor splice sites.

Top-k splice site accuracy evaluation is computed as follows: For a gene sequence of length $L$, containing 10 true donor splice sites ($n_{true}^{donor} = 10$) and 10 true acceptor splice sites ($n_{true}^{acceptor} = 10$), the model generates probability scores for donor and acceptor sites across the sequence. For top-k accuracy, the $k \times n_{true}^{class}$ highest-scoring predictions are extracted per class (donor/acceptor), where $n_{true}^{class} = 10$ for each splice site type. Class-specific accuracy is calculated as follows (*Equation 17*):

$$Top - k^{class} = \frac{\sum_{i=1}^{k \times n_{true}^{class}} I\left(p_i^{class} \in S_{true}^{class}\right)}{k \times n_{true}^{class}}, class \in \{donor, acceptor\} \tag{17}$$

where $p_{true}^{class}$ denotes the $i$th ranked prediction for the class, $S_{true}^{class}$ is the set of true splice sites for the class, and $\mathrm{I}$ is an indicator function.

For $k = 1$, this evaluates whether true splice sites are present among the top 10 donor and top 10 acceptor predictions (20 total). Accuracy is defined as the proportion of true sites correctly identified within this subset. For $k = 2$, the evaluation expands to the top 20 predictions per class (40 total). This metric quantifies the model's ability to prioritize true splice sites within ranked candidate positions.

### Accuracy, precision, recall, F1-score, and AUPRC

Unlike top-k accuracy, which requires knowing the number of ground-truth splice sites, we can determine the predicted class for each position based on the highest probability across each dimension and evaluate predictions as true positives (TP), true negatives (TN), false positives (FP), or false negatives (FN) based on a preset threshold. After labeling each site in a given sequence, we calculate accuracy, precision, recall, F1-score, and AUPRC with the threshold 0.5 for the sequence prediction. For example, for donor sites, the metrics are calculated as follows (*Equations 18–20*):

$$Accuracy_{donor\ site} = \frac{TP_{donor\ site} + FP_{donor\ site}}{TP_{donor\ site} + FP_{donor\ site} + TN_{donor\ site} + FN_{donor\ site}} \tag{18}$$

$$Precision_{donor\ site} = \frac{TP_{donor\ site}}{TP_{donor\ site} + FN_{donor\ site}} \tag{19}$$

$$Recall_{donor\ site} = \frac{TP_{donor\ site}}{TP_{donor\ site} + TN_{donor\ site}} \tag{20}$$

### Benchmarking SpliceAI-Keras and OSAI_{MANE}

To compare OSAI_{MANE} with SpliceAI (*Jaganathan et al., 2019*), we benchmarked the computational efficiency and performance of both tools across various metrics in two tasks: large-scale prediction (using the 'predict' module) and variant effect prediction (using the 'variant' module). The results are shown in *Figure 2G and H*.

### Time and CPU/GPU memory profiling

The SpliceAI-Keras and OSAI_{MANE} were benchmarked using the Scalene profiling tool (https://github.com/plasma-umass/scalene; *Berger, 2025*), a Python-specific profiler which handles CPU, GPU, and memory profiling and evaluates code line-by-line. We measured the following metrics: elapsed CPU time, peak CPU memory, peak GPU memory, percentage of CPU time in low-level C code, CPU memory growth rate, and average memory usage. The first three metrics are visualized in *Figure 2E*, while the remaining are presented in *Figure 2—figure supplement 6*.

### Predict benchmark design

The objective of this benchmark was to provide a fair comparison of the computational efficiency of SpliceAI and OSAI_{MANE}, with the key difference being that SpliceAI is implemented in Keras (https://github.com/keras-team/keras; *Chollet, 2025*) and OSAI_{MANE} is implemented in PyTorch (https://github.com/pytorch/pytorch; *Chintala et al., 2025*). While SpliceAI includes a variant effect prediction utility, it lacks a dedicated tool for large-scale predictions. To address this, we extracted the core Keras-wrapped prediction code from SpliceAI's variant tool and integrated it into our 'predict' utility, which has our specific data preprocessing and BED file generation. We call this tool 'SpliceAI-Keras'.

For this experiment, we randomly sampled 1000 protein-coding genes from MANE and benchmarked the tools on increasingly large subsets of the genes. The prediction task is to identify all splice sites within the gene locus.

With the Scalene profiler active, we ran both SpliceAI-Keras and OSAI_{MANE} with the 'predict' subcommand across all five models (per default usage), extracting the averaged predictions, and repeated the process for a total of 5 trials. We further benchmarked every model size (80, 400, 2000, and 10,000 bp flanking size). The graph visualizations depict the mean metrics as a solid line, while the shaded region represents the variance between trials.

If the computation encountered an out-of-memory error that caused the prediction to stall, we discarded that trial. Note that running Scalene also requires dedicated memory usage, which was not

included in the graphs. Some input sizes yielded no successful trials for SpliceAI-Keras, reflected in missing datapoints.

## Variant benchmark design

We compared SpliceAI 'variant' tool https://github.com/Illumina/SpliceAI (*McRae and Jaganathan, 2025*) with OpenSpliceAI 'variant' command (in default mode) in an analogous manner to the 'predict' benchmark. This experiment compared every model size across both SpliceAI-Keras and OSAI$_{MANE}$ for 5 trials. For the input VCF file, we used the Mills and 1000 Genomes Project gold standard dataset of known indels in GRCh38, provided by the Broad Institute. We randomly sampled 1000 indels and benchmarked on increasingly large subsets of this data. The resultant graph is shown in *Figure 2— figure supplement 6G–L*.

## ISM analysis

The ISM study compares the prediction patterns of OSAI$_{MANE}$ with SpliceAI (*Jaganathan et al., 2019*) to demonstrate their similarity and biological relevance. The ISM experiments investigate the effect of mutations on predicted splicing patterns, and we replicate several key experiments from *Jaganathan et al., 2019*, to illustrate this.

## Importance score

For assessing the impact of a mutation in a given base position on the strength of the splice site, we calculate an 'importance score' as follows (*Equation 21*):

$$Importance = S_{ref} - \frac{S_A + S_C + S_G + S_T}{4} \tag{21}$$

where $S_{ref}$ denotes the splice site score of the wild-type sequence, and $S_A$, $S_C$, $S_G$, and $S_T$ denote the splice site's score with each of the corresponding base substitutions. The importance score reflects the decrease in the predicted strength of the splice site when the target nucleotide is mutated and can be regarded as the significance of the target base in contributing toward the activation of the splice site (*Equation 20*).

## Single-nucleotide variation in short exons

Pursuant to the protocol of *Jaganathan et al., 2019*, we investigate the importance of various base positions around two known short exons in the human genome, U2SURP exon 9 and DST exon 2. We evaluate the scores using both SpliceAI-Keras and OSAI$_{MANE}$ to compare the predicted splicing patterns. The findings are summarized in *Figure 6A*.

For each example, we extract the entire exon, as well as part of the intronic region upstream and downstream of the exon to illustrate the full range of base positions. We then mutate each base of the input sequence to the three possible single nucleotide variations at that position. For the reference sequence and each mutated sequence, we collect the acceptor site score and calculate an importance score for that position. We finally visualize the acceptor site importance scores, corresponding to the vertical size of the DNA logo of the reference sequence.

Note that for the U2SURP investigation, we use the same input sequence as *Jaganathan et al., 2019*, which is extracted from the hg19 assembly, to ensure reproducibility. The DST exon used is updated with the more recent GRCh38 assembly. DST was selected to ensure representation across the train and test datasets. Additionally, we note that the DST gene is on the reverse strand, but we display the forward strand (the strand that the splicing models use as input) in our visualization for ease of comparison. Lastly, for this experiment and all subsequent ISM studies, we extract an additional 10,000 bp around the input sequence (as opposed to N-padding) and use the SpliceAI-10k and OSAI$_{MANE}$-10k models to perform prediction.

## Concordance evaluation of ISM importance scores between OSAI$_{MANE}$ and SpliceAI

To assess agreement between OSAI$_{MANE}$ and SpliceAI across a broad set of splice sites, we applied our ISM procedure to 100 randomly chosen donor sites and 100 randomly chosen acceptor sites. For each

site, we extracted a 5001 nt window centered on the annotated splice junction and, at every coordinate within that window, substituted the reference base with each of the three alternative nucleotides. We recorded the change in predicted splice site probability for each mutation and then averaged these Δ-scores at each position to produce a 5001-score ISM importance profile per site.

Next, for each splice site, we computed the Pearson correlation coefficient between the paired importance profiles from ensembled OSAI$_{MANE}$ and ensembled SpliceAI. The median correlation was 0.857 for all splice sites. Ten additional zoom-in representative splice site DNA logo comparisons are provided in *Figure 6—figure supplement 1*.

## Batch mutagenesis for donor and acceptor site motif recognition

To establish a more representative idea of the splicing pattern around splice sites, we scaled up the ISM experiment across multiple donor and acceptor sites. The data consists of 100 randomly sampled donor and 100 randomly sampled acceptor sites from the testing dataset (chromosomes 1, 3, 5, 7, and 9). Each sequence is 400 bp with the donor and acceptor splicing motifs located at the midpoint of the sequence.

For each position in the sequence, we mutated the base to every other base and measured the decrease in the strength of the central splice site score. Again, the measurement was taken as the average across all five models for both SpliceAI-Keras and OSAI$_{MANE}$ (10k model size). We considered each point mutation separately here. We repeated this for every sequence and took the average result across the 100 samples.

Finally, we displayed the averaged scores in a DNA logo (*Figure 6B*), where the vertical size of a base denotes the magnitude of decrease in splice site strength when the original base at that position is mutated to that base. We show the central 80 base positions in our visualization for readability.

We note that SpliceAI-Keras runs many orders of magnitude slower than OSAI$_{MANE}$. In order to speed up the computation, we split up the input into 10 smaller batches of 10 transcripts each and processed them in parallel on a GPU cluster. The results were aggregated, then averaged.

## Cryptic splicing mutation analysis

To reproduce another key experiment from *Jaganathan et al., 2019* (*Figure 2A*), we selected a specific G to A point mutation in intron 14 of the MYBPC3 gene which results in a cryptic splicing variant. We further select another example of a cryptic splicing variant that is validated in vitro, from *Qian et al., 2021*. This A to G point mutation occurs deep in intron 16 of OPA1 and was shown to have the highest aberrant-only splicing in minigene assays, with the inclusion of an entire cryptic exon upstream of the mutation.

For each variant, we scored the wild-type sequence and the mutated sequence and calculated the change in donor and acceptor scores for all bases around the mutation (specifically at the expected locations of cryptic splice site gain), for both SpliceAI-Keras and OSAI$_{MANE}$. We visualized the most significant splice site gain or loss events, along with their raw splice site scores, in *Figure 6C*.

Because the sequence near the acceptor site differs in the MYBPC3 gene between the hg19 assembly (used in *Jaganathan et al., 2019*) and GRCh38 assembly, we opted to retain the hg19 annotation for this example, for the purpose of reproducing SpliceAI's results.

## Full prediction of the CFTR gene

We finally reproduce the full prediction of splice sites in the CFTR gene using both SpliceAI-Keras and OSAI$_{MANE}$. Notably different, however, is the fact that we are using the updated GRCh38 assembly, and that we use a fixed score threshold of 0.5 (default for 'predict') to identify splice sites.

We extracted the full CFTR gene and ran both tools, taking the averaged score across all five models for each tool. The findings are summarized in *Figure 6D*. The exon plot displays the reference MANE annotation with the locations of the predicted donor and acceptor sites marked in color. The histogram below visualizes the corresponding donor and acceptor score distributions.

We further note that we extract the true 10k DNA flanking sequence around the gene (as opposed to N-padding), which is why our results do not appear to exactly replicate the original study from *Jaganathan et al., 2019*.

## Acknowledgements

We thank all the members of the Salzberg and Pertea Labs for their valuable discussions and insights. This research was supported in part by the U.S. National Institutes of Health under grants R01-HG006677, R35-GM130151, and R35-GM156470, and by the U.S. National Science Foundation under DBI 2412449. Computational work was carried out at the Advanced Research Computing at Hopkins (ARCH) core facility, supported in part by NSF grant OAC 1920103.

## Additional information

### Funding

| Funder | Grant reference number | Author |
| --- | --- | --- |
| U.S. National Institute of Health | R01-HG006677 | Kuan-Hao Chao<br>Alan Mao<br>Steven L Salzberg |
| U.S. National Institute of Health | R35-GM130151 | Kuan-Hao Chao<br>Alan Mao<br>Steven L Salzberg |
| U.S. National Institute of Health | R35-GM156470 | Kuan-Hao Chao<br>Alan Mao<br>Mihaela Pertea |
| National Science Foundation | DBI 2412449 | Kuan-Hao Chao<br>Alan Mao<br>Mihaela Pertea |
| National Science Foundation | OAC 1920103 | Kuan-Hao Chao<br>Alan Mao<br>Steven L Salzberg<br>Mihaela Pertea |

The funders had no role in study design, data collection and interpretation, or the decision to submit the work for publication.

### Author contributions

Kuan-Hao Chao, Conceptualization, Resources, Software, Formal analysis, Validation, Visualization, Methodology, Writing – original draft, Writing – review and editing; Alan Mao, Software, Formal analysis, Visualization, Methodology, Writing – original draft, Writing – review and editing; Anqi Liu, Formal analysis, Methodology, Writing – original draft, Writing – review and editing; Steven L Salzberg, Mihaela Pertea, Conceptualization, Supervision, Funding acquisition, Validation, Investigation, Writing – original draft, Project administration, Writing – review and editing

### Author ORCIDs

Kuan-Hao Chao ⓘ https://orcid.org/0000-0003-0099-0692
Alan Mao ⓘ http://orcid.org/0000-0003-2381-0607
Steven L Salzberg ⓘ https://orcid.org/0000-0002-8859-7432
Mihaela Pertea ⓘ https://orcid.org/0000-0003-0762-8637

Reviewer #1 (Public review): https://doi.org/10.7554/eLife.107454.3.sa1
Reviewer #2 (Public review): https://doi.org/10.7554/eLife.107454.3.sa2
Author response https://doi.org/10.7554/eLife.107454.3.sa3

## Additional files

### Supplementary files

MDAR checklist

## Data availability

OpenSpliceAI is implemented as a Python package, using Pytorch framework. OpenSpliceAI project is freely available on GitHub at: https://github.com/Kuanhao-Chao/OpenSpliceAI (*Chao, 2025*) and is available on PyPi: https://pypi.org/project/openspliceai/. The OpenSpliceAI documentation is available at: https://ccb.jhu.edu/openspliceai/.

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
