## [Editor Report · eLife Assessment]

This **valuable** study introduces a modern and accessible PyTorch reimplementation of the widely used SpliceAI model for splice site prediction. The authors provide **convincing** evidence that their OpenSpliceAI implementation matches the performance of the original while improving usability and enabling flexible retraining across species. These advances are likely to be of broad interest to the computational genomics community.

---

## [Referee Report · Reviewer #1 (Public review)]

Summary:

Chao et al. produced an updated version of the SpliceAI package using modern deep learning frameworks. This includes data preprocessing, model training, direct prediction, and variant effect prediction scripts. They also added functionality for model fine-tuning and model calibration. They convincingly evaluate their newly trained models against those from the original SpliceAI package and investigate how to extend SpliceAI to make predictions in new species. Their comparisons to the original SpliceAI models are convincing on the grounds of model performance and their evaluation of how well the new models match the original's understanding of non-local mutation effects. However, their evaluation of the new calibration functionality would benefit from a more nuanced discussion of the limitations of calibration.

Strengths

(1) They provide convincing evidence that their new implementation of SpliceAI matches the performance and mutation effect estimation capabilities of the original model on a similar dataset while benefiting from improved computational efficiencies. This will enable faster prediction and retraining of splicing models for new species as well as easier integration with other modern deep learning tools.

(2) They produce models with strong performance on non-human model species and a simple well well-documented pipeline for producing models tuned for any species of interest. This will be a boon for researchers working on splicing in these species and make it easy for researchers working on new species to generate their own models.

(3) Their documentation is clear and abundant. This will greatly aid the ability of others to work with their code base.

Weaknesses

(1) Their discussion of their package's calibration functionality does not adequately acknowledge the limitations of model calibration. This is problematic as this is a package intended for general use and users who are not experienced in modeling broadly and the subfield of model calibration specifically may not already understand these limitations. This could lead to serious errors and misunderstandings down the road. A model is not calibrated or uncalibrated in and of itself, only with respect to a specific dataset. In this case they calibrated with respect to the training dataset, a set of canonical transcript annotations. This is a perfectly valid and reasonable dataset to calibrate against. However, this is unlikely to be the dataset the model is applied to in any downstream use case, and this calibration is not guaranteed or expected to hold for any shift in the dataset distribution. For example, in the next section they use ISM based approaches to evaluate which sequence elements the model is sensitive to and their calibration would not be expected to hold for this set of predictions. This issue is particularly worrying in the case of their model because annotation of canonical transcript splice sites is a task that it is unlikely their model will be applied to after training. Much more likely tasks will be things such as predicting the effects of mutations, identification of splice sites that may be used across isoforms beyond just the canonical one, identification of regulatory sequences through ISM, or evaluation of human created sequences for design or evaluation purposes (such as in the context of an MPSA or designing a gene to splice a particular way), we would not expect their calibration to hold in any of these contexts. To resolve this issue, the authors should clarify and discuss this limitation in their paper (and in the relevant sections of the package documentation) to avoid confusing downstream users.

(2) The clarity of their analysis of mutation effects could be improved with some minor adjustments. While they report median ISM importance correlation it would be helpful to see a histogram of the correlations they observed. Instead of displaying (and calculating correlations using) importance scores of only the reference sequence, showing the importance scores for each nucleotide at each position provides a more informative representation. This would also likely make the plots in 6B clearer.

---

## [Referee Report · Reviewer #2 (Public review)]

Summary:

The paper by Chao et al offers a reimplantation of the SpliceAI algorithm in PyTorch so that the model can more easily/efficiently be retrained. They apply their new implementation of the SpliceAI algorithm, which they call OpenSpliceAI, to several species and compare it against the original model, showing that the results are very similar and that in some small species pre-training on other species helps improve performance.

Strengths:

On the upside, the code runs fine and it is well documented.

Weaknesses:

The paper itself does not offer much beyond reimplementing SpliceAI. There is no new algorithm, new analysis, new data, or new insights into RNA splicing. There is not even any comparison to many of the alternative methods that have since been published to surpass SpliceAI. Given that some of the authors are well known with a long history of important contributions, our expectations were admittedly different. Still, we hope some readers will find the new implementation useful.

Update for the revised version:

The update includes mostly clarifications for tech questions/comments raised by the other two reviewers. There is no additional analysis/results that changes our above initial assessment of this paper's contribution.

---

## [Author Response]

The following is the authors’ response to the original reviews.

**Reviewer #1 (Public review):**
Summary:Chao et al. produced an updated version of the SpliceAI package using modern deep learning frameworks. This includes data preprocessing, model training, direct prediction, and variant effect prediction scripts. They also added functionality for model fine-tuning and model calibration. They convincingly evaluate their newly trained models against those from the original SpliceAI package and investigate how to extend SpliceAI to make predictions in new species. While their comparisons to the original SpliceAI models are convincing on the grounds of model performance, their evaluation of how well the new models match the original's understanding of non-local mutation effects is incomplete. Further, their evaluation of the new calibration functionality would benefit from a more nuanced discussion of what set of splice sites their calibration is expected to hold for, and tests in a context for which calibration is needed.Strengths:(1) They provide convincing evidence that their new implementation of SpliceAI matches the performance of the original model on a similar dataset while benefiting from improved computational efficiencies. This will enable faster prediction and retraining of splicing models for new species as well as easier integration with other modern deep learning tools.(2) They produce models with strong performance on non-human model species and a simple, well-documented pipeline for producing models tuned for any species of interest. This will be a boon for researchers working on splicing in these species and make it easy for researchers working on new species to generate their own models.(3) Their documentation is clear and abundant. This will greatly aid the ability of others to work with their code base.

We thank the reviewer for these positive comments.

Weaknesses:(1) The authors' assessment of how much their model retains SpliceAI's understanding of "nonlocal effects of genomic mutations on splice site location and strength" (Figure 6) is not sufficiently supported. Demonstrating this would require showing that for a large number of (non-local) mutations, their model shows the same change in predictions as SpliceAI or that attribution maps for their model and SpliceAI are concordant even at distances from the splice site. Figure 6A comes close to demonstrating this, but only provides anecdotal evidence as it is limited to 2 loci. This could be overcome by summarizing the concordance between ISM maps for the two models and then comparing across many loci. Figure 6B also comes close, but falls short because instead of comparing splicing prediction differences between the models as a function of variants, it compares the average prediction difference as a function of the distance from the splice site. This limits it to only detecting differences in the model's understanding of the local splice site motif sequences. This could be overcome by looking at comparisons between differences in predictions with mutants directly and considering non-local mutants that cause differences in splicing predictions.

We agree that two loci are insufficient to demonstrate preservation of non-local effects. To address this, we have extended our analysis to a larger set of sites: we randomly sampled 100 donor and 100 acceptor sites, applied our ISM procedure over a 5,001 nt window centered at each site for both models, and computed the ISM map as before. We then calculated the Pearson correlation between the collection of OSAI_MANE_ and SpliceAI ISM importance scores. We also created 10 additional ISM maps similar to those in Figure 6A, which are now provided in Figure S23.

Follow is the revised paragraph in the manuscript’s Results section:

First, we recreated the experiment from Jaganathan et al. in which they mutated every base in a window around exon 9 of the U2SURP gene and calculated its impact on the predicted probability of the acceptor site. We repeated this experiment on exon 2 of the DST gene, again using both SpliceAI and OSAI_MANE_ . In both cases, we found a strong similarity between the resultant patterns between SpliceAI and OSAI_MANE_, as shown in Figure 6A. To evaluate concordance more broadly, we randomly selected 100 donor and 100 acceptor sites and performed the same ISM experiment on each site. The Pearson correlation between SpliceAI and OSAI_MANE_ yielded an overall median correlation of 0.857 (see Methods; additional DNA logos in Figure S23).

To characterize the local sequence features that both models focus on, we computed the average decrease in predicted splice-site probability resulting from each of the three possible singlenucleotide substitutions at every position within 80bp for 100 donor and 100 acceptor sites randomly sampled from the test set (Chromosomes 1, 3, 5, 7, and 9). Figure 6B shows the average decrease in splice site strength for each mutation in the format of a DNA logo, for both tools.

We added the following text to the Methods section:

Concordance evaluation of ISM importance scores between OSAI_MANE_ and SpliceAI

To assess agreement between OSAI_MANE_ and SpliceAI across a broad set of splice sites, we applied our ISM procedure to 100 randomly chosen donor sites and 100 randomly chosen acceptor sites. For each site, we extracted a 5,001 nt window centered on the annotated splice junction and, at every coordinate within that window, substituted the reference base with each of the three alternative nucleotides. We recorded the change in predicted splice-site probability for each mutation and then averaged these Δ-scores at each position to produce a 5,001-score ISM importance profile per site.

Next, for each splice site we computed the Pearson correlation coefficient between the paired importance profiles from ensembled OSAI_MANE_ and ensembled SpliceAI. The median correlation was 0.857 for all splice sites. Ten additional zoom-in representative splice site DNA logo comparisons are provided in Supplementary Figure S23.

(2) The utility of the calibration method described is unclear. When thinking about a calibrated model for splicing, the expectation would be that the models' predicted splicing probabilities would match the true probabilities that positions with that level of prediction confidence are splice sites. However, the actual calibration that they perform only considers positions as splice sites if they are splice sites in the longest isoform of the gene included in the MANE annotation. In other words, they calibrate the model such that the model's predicted splicing probabilities match the probability that a position with that level of confidence is a splice site in one particular isoform for each gene, not the probability that it is a splice site more broadly. Their level of calibration on this set of splice sites may very well not hold to broader sets of splice sites, such as sites from all annotated isoforms, sites that are commonly used in cryptic splicing, or poised sites that can be activated by a variant. This is a particularly important point as much of the utility of SpliceAI comes from its ability to issue variant effect predictions, and they have not demonstrated that this calibration holds in the context of variants. This section could be improved by expanding and clarifying the discussion of what set of splice sites they have demonstrated calibration on, what it means to calibrate against this set of splice sites, and how this calibration is expected to hold or not for other interesting sets of splice sites. Alternatively, or in addition, they could demonstrate how well their calibration holds on different sets of splice sites or show the effect of calibrating their models against different potentially interesting sets of splice sites and discuss how the results do or do not differ.

We thank the reviewer for highlighting the need to clarify our calibration procedure. Both SpliceAI and OpenSpliceAI are trained on a single “canonical” transcript per gene: SpliceAI on the hg 19 Ensembl/Gencode canonical set and OpenSpliceAI on the MANE transcript set. To calibrate each model, we applied post-hoc temperature scaling, i.e. a single learnable parameter that rescales the logits before the softmax. This adjustment does not alter the model’s ranking or discrimination (AUC/precision–recall) but simply aligns the predicted probabilities for donor, acceptor, and non-splice classes with their observed frequencies. As shown in our reliability diagrams (Fig. S16-S22), temperature scaling yields negligible changes in performance, confirming that both SpliceAI and OpenSpliceAI were already well-calibrated. However, we acknowledge that we didn’t measure how calibration might affect predictions on non-canonical splice sites or on cryptic splicing. It is possible that calibration might have a detrimental effect on those, but because this is not a key claim of our paper, we decided not to do further experiments. We have updated the manuscript to acknowledge this potential shortcoming; please see the revised paragraph in our next response.

(3) It is difficult to assess how well their calibration method works in general because their original models are already well calibrated, so their calibration method finds temperatures very close to 1 and only produces very small and hard to assess changes in calibration metrics. This makes it very hard to distinguish if the calibration method works, as it doesn't really produce any changes. It would be helpful to demonstrate the calibration method on a model that requires calibration or on a dataset for which the current model is not well calibrated, so that the impact of the calibration method could be observed.

It’s true that the models we calibrated didn’t need many changes. It is possible that the calibration methods we used (which were not ours, but which were described in earlier publications) can’t improve the models much. We toned down our comments about this procedure, as follows.

Original:

“Collectively, these results demonstrate that OSAIs were already well-calibrated, and this consistency across species underscores the robustness of OpenSpliceAI’s training approach in diverse genomic contexts.”

Revised:

“We observed very small changes after calibration across phylogenetically diverse species, suggesting that OpenSpliceAI’s training regimen yielded well‐calibrated models, although it is possible that a different calibration algorithm might produce further improvements in performance.”

**Reviewer #2 (Public review):**
Summary:The paper by Chao et al offers a reimplementation of the SpliceAI algorithm in PyTorch so that the model can more easily/efficiently be retrained. They apply their new implementation of the SpliceAI algorithm, which they call OpenSpliceAI, to several species and compare it against the original model, showing that the results are very similar and that in some small species, pretraining on other species helps improve performance.Strengths:On the upside, the code runs fine, and it is well documented.Weaknesses:The paper itself does not offer much beyond reimplementing SpliceAI. There is no new algorithm, new analysis, new data, or new insights into RNA splicing. There is no comparison to many of the alternative methods that have since been published to surpass SpliceAI. Given that some of the authors are well-known with a long history of important contributions, our expectations were admittedly different. Still, we hope some readers will find the new implementation useful.

We thank the reviewer for the feedback. We have clarified that OpenSpliceAI is an open-source PyTorch reimplementation optimized for efficient retraining and transfer learning, designed to analyze cross-species performance gains, and supported by a thorough benchmark and the release of several pretrained models to clearly position our contribution.

**Reviewer #3 (Public review):**
Summary:The authors present OpenSpliceAI, a PyTorch-based reimplementation of the well-known SpliceAI deep learning model for splicing prediction. The core architecture remains unchanged, but the reimplementation demonstrates convincing improvements in usability, runtime performance, and potential for cross-species application.Strengths:The improvements are well-supported by comparative benchmarks, and the work is valuable given its strong potential to broaden the adoption of splicing prediction tools across computational and experimental biology communities.Major comments:Can fine-tuning also be used to improve prediction for human splicing? Specifically, are models trained on other species and then fine-tuned with human data able to perform better on human splicing prediction? This would enhance the model's utility for more users, and ideally, such fine-tuned models should be made available.

We evaluated transfer learning by fine-tuning models pretrained on mouse (OSAI_Mouse_), honeybee (OSAI_Honeybee_), Arabidopsis (OSAI_Arabidopsis_), and zebrafish (OSAI_Zebrafish_) on human data. While transfer learning accelerated convergence compared to training from scratch, the final human splicing prediction accuracy was comparable between fine-tuned and scratch-trained models, suggesting that performance on our current human dataset is nearing saturation under this architecture.

We added the following paragraph to the Discussion section:

We also evaluated pretraining on mouse (OSAI_Mouse_), honeybee (OSAI_Honeybee_), zebrafish (OSAI_Zebrafish_), and Arabidopsis (OSAI_Arabidopsis_) followed by fine-tuning on the human MANE dataset. While cross-species pretraining substantially accelerated convergence during fine-tuning, the final human splicing-prediction accuracy was comparable to that of a model trained from scratch on human data. This result indicates that our architecture seems to capture all relevant splicing features from human training data alone, and thus gains little or no benefit from crossspecies transfer learning in this context (see Figure S24).

**Reviewer #1 (Recommendations for the authors):**

We thank the editor for summarizing the points raised by each reviewer. Below is our point-bypoint response to each comment:

(1) In Figure 3 (and generally in the other figures) OpenSpliceAI should be replaced with OSAI_{Training dataset} because otherwise it is hard to tell which precise model is being compared. And in Figure 3 it is especially important to emphasize that you are comparing a SpliceAI model trained on Human data to an OSAI model trained and evaluated on a different species.

We have updated the labels in Figures 3, replacing “OpenSpliceAI” with “OSAI_{training dataset}” to more clearly specify which model is being compared.

(2) Are genes paralogous to training set genes removed from the validation set as well as the test set? If you are worried about data leakage in the test set, it makes sense to also consider validation set leakage.

Thank you for this helpful suggestion. We fully agree, and to avoid any data leakage we implemented the identical filtering pipeline for both validation and test sets: we excluded all sequences paralogous or homologous to sequences in the training set, and further removed any sequence sharing > 80 % length overlap and > 80 % sequence identity with training sequences. The effect of this filtering on the validation set is summarized in Supplementary Figure S7C.

**Reviewer #3 (Recommendations for the authors):**
(1) The legend in Figure 3 is somewhat confusing. The labels like "SpliceAI-Keras (species name)" may imply that the model was retrained using data from that species, but that's not the case, correct?

Yes, “SpliceAI-Keras (species name)” was not retrained; it refers to the released SpliceAI model evaluated on the specified species dataset. We have revised the Figure 3 legends, changing “SpliceAI-Keras (species name)” to “SpliceAI-Keras” to clarify this.

(2) Please address the minor issues with the code, including ensuring the conda install works across various systems.

We have addressed the issues you mentioned. OpenSpliceAI is now available on Conda and can be installed with: conda install openspliceai.

The conda package homepage is at: https://anaconda.org/khchao/openspliceai We’ve also corrected all broken links in the documentation.

(3) Utility:I followed all the steps in the Quick Start Guide, and aside from the issues mentioned below, everything worked as expected.I attempted installation using conda as described in the instructions, but it was unsuccessful. I assume this method is not yet supported.In Quick Start Guide: predict, the link labeled "GitHub (models/spliceai-mane/10000nt/)" appears to be incorrect. The correct path is likely "GitHub (models/openspliceaimane/10000nt/)".In Quick Start Guide: variant (https://ccb.jhu.edu/openspliceai/content/quick_start_guide/quickstart_variant.html#quick-startvariant), some of the download links for input files were broken. While I was able to find some files in the GitHub repository, I think the -A option should point to data/grch37.txt, not examples/data/input.vcf, and the -I option should be examples/data/input.vcf, not data/vcf/input.vcf.

Thank you for catching these issues. We’ve now addressed all issues concerning Conda installation and file links. We thank the editor for thoroughly testing our code and reviewing the documentation.